# Solar overall water-splitting by a spin-hybrid all-organic semiconductor

Xinyu Lin[1], Yue Hao[1], Yanjun Gong[2,3], Peng Zhou[4], Dongge Ma [5], Zhonghuan Liu[1], Yuming Sun[1], Hongyang Sun[1], Yahui Chen[1], Shuhan Jia[1], Wanhe Li[1], Chengqi Guo[1], Yiying Zhou[1], Pengwei Huo [1], Yan Yan [1]✉, Wanhong Ma [2]✉, Shouqi Yuan [1]✉ & Jincai Zhao [2]

Direct solar-to-hydrogen conversion from pure water using all-organic heterogeneous catalysts remains elusive. The challenges are twofold: (i) full-band low-frequent photons in the solar spectrum cannot be harnessed into a unified $S_1$ excited state for water-splitting based on the common Kasha-allowed $S_O \to S_1$ excitation; (ii) the $H^+ \to H_2$ evolution suffers the high overpotential on pristine organic surfaces. Here, we report an organic molecular crystal nanobelt through the self-assembly of spin-one open-shell perylene diimide diradical anions (:PDI$^{2-}$) and their tautomeric spin-zero closed-shell quinoid isomers (PDI$^{2-}$). The self-assembled :PDI$^{2-}$/PDI$^{2-}$ crystal nanobelt alters the spin-dependent excitation evolution, leading to spin-allowed $S_OS_1 \to {}^1(TT) \to T_1 + T_1$ singlet fission under visible-light (420 nm-700 nm) and a spin-forbidden $S_O \to T_1$ transition under near-infrared (700 nm-1100 nm) within spin-hybrid chromophores. With a triplet-triplet annihilation upconversion, a newly formed $S_1$ excited state on the diradical-quinoid hybrid induces the $H^+$ reduction through a favorable hydrophilic diradical-mediated electron transfer, which enables simultaneous $H_2$ and $O_2$ production from pure water with an average apparent quantum yield over 1.5% under the visible to near-infrared solar spectrum.

Extracting hydrogen ($H_2$) from pure water powered by clean and inexhaustible sunlight is a long-dreamed solar energy conversion strategy in a way that compensates for the intermittency of solar irradiation[1,2]. Such technology requires a photocatalyst of superior visible-to-NIR light response that is maximally close to the solar spectrum, long-enough exciton lifetimes for ns-scale water splitting, and proper band structure for both hydrogen and oxygen evolution reactions (i.e., HER and OER)[3,4]. Although much desired, current-developed semiconductor catalysts, including transition-metal oxides[5], -oxynitride[6], metal-organic hybrid framework-based materials[7], and metal-free inorganic semiconductors (e. g., $C_3N_4$ and

BN)[8,9], cannot meet all the requirements for commercial application with >10% solar-to-hydrogen (STH) conversion efficiency[10,11]. Besides the commonly low apparent quantum yields (AQYs) at the wavelength ≥420 nm visible-to-NIR region of incident sunlight[4], current stereotypical photocatalyst designs require the loading of expensive or poisonous noble-metal/metal-oxide as co-catalysts to help release $H_2$ and $O_2$[1–4]. In this sense, metal-free all-organic catalysts capable of solar overall water-splitting hold research perspectives[11].

Compared with metal-based inorganic or organic-inorganic complex photocatalysts, covalent C-H-X organic materials available for various structural tailor and functional designs show superior

[1]School of Chemistry & Chemical Engineering/Research Center of Fluid Machinery Engineering and Technology, Jiangsu University, Zhenjiang 212013, China. [2]Key Laboratory of Photochemistry, Institute of chemistry, Chinese Academy of Sciences, 100190 Beijing, China. [3]University of Chinese Academy of Sciences, 100049 Beijing, China. [4]Electrical Engineering & Computer Science, University of Michigan, Ann Arbor, MI 48109-2122, USA. [5]Department of Chemistry, College of Chemistry and Materials Engineering, Beijing Technology and Business University, 100048 Beijing, China. ✉e-mail: dgy5212004@163.com; whma@iccas.ac.cn; shouqiy@ujs.edu.cn

advantages in light-harvesting initialed by π → π or n → π transitions[11]. Some even reach the world record for HER half-reaction efficiency over inorganic catalysts[12]. However, due to the fast radiative recombination of direct $S_O \rightarrow S_1$ excitations[13,14] and the thermodynamical energy requirement of overall water-splitting[4], direct STH conversion over common organic materials is impossible with low-frequency photons around 420–1000 nm, ~50% energy of the incident solar spectrum[15]. As a solution, the upconversion strategy based on the triplet-fusion can double the incident photon energy, driving the water-splitting reaction that is thermodynamically impossible to direct low-frequency irradiation[16]. Compared to the Kasha-allowed singlet excitation ($S_O \rightarrow S_1$), the triplet-fusion pathway is mediated by a long-lived triplet-triplet (*TT*) exciton pair[16], which is inaccessible by a direct $S_O \rightarrow T_1$ transition due to the disallowed spin[13]. Consequently, dark-state *TT* pairs of an organic chromophore in a sensitizer-annihilator system can only be indirectly populated either by the $S_1 \rightarrow T_n$ intersystem crossing (ISC, n ≥ 1) (Fig. 1a) or the $S_O S_1 \rightarrow {}^1(TT) \rightarrow (TT)$ singlet fission (SF) process[14,17,18], which generally requires strong UV/vis absorption to produce the initial singlet excited state on the sensitizer and a robust conjugation of the sensitizer-annihilator chromophore pair[19]. This is incompatible with utilizing full-band low-frequency photons in the solar spectrum. Moreover, as a competing pathway to the Dexter triplet-triplet energy transfer (TTET), high-energy triplet excitons arose from little $S_1/T_1$ energy difference are easily quenched by triplet $O_2$[20], the main product of the water-splitting reaction ($H_2O \rightarrow H_2 + O_2$), to produce singlet $O_2$, rendering STH conversions unsustainable. In addition, due to the common high overpotential on simple hydrocarbon surfaces, the $H^+ \rightarrow H_2$ evolution reaction (HER), a key half-reaction in water splitting, can hardly occur on organic catalysts unless loaded with noble-/transition-metal-based cocatalysts.

In contrast, a triplet-fusion pathway with *TT* pairs from intrinsic $S_O \rightarrow T_1$ transition can harness a wide range of incident sunlight, including near-infrared (NIR) photon energy, and be immune to $O_2$ quenching with low triplet energy levels. Until now, such a spin-forbidden $S_O \rightarrow T_n$ transition, however, has only been achieved in a few examples by spin-orbit coupling with heavy atoms (e.g., Lanthanide,

Iodine, Osmium, Platinum, and Gold)[21–25]. Although much desired, all-organic material-induced direct $S_O \rightarrow T_n$ transition without any heavy atoms has yet to be identified. We envision the direct $S_O \rightarrow T_1$ transition by a robust spin-orbit coupling between the preset open-shell triplet chromophore (i.e., diradical) and the singlet closed-shell isomer chromophore within a heavily conjugated crystal. Such a design not only makes the full use of the visible-to-NIR solar spectrum by the two-band excitation but also creates a proton-favorable electron transfer channel by diradical-dominated reactive interface on all-organic catalysts to break through the HER bottleneck (see Fig. 1b).

## Results

### Synthesis and catalyst structure

We opt N, N′-di(propanoic acid)-perylene-3, 4, 9, 10-tetracarboxylic diimide (denoted PDI throughout the article) (Fig. 2g) as the model perylene diimide parent to prepare the target spin-hybrid all-organic semiconductor (for details of the preparation methods, see Method). The pristine PDI in aqueous solution exhibits broad visible-light absorption with double peaks at 460 nm and 596 nm (Fig. 2a). After the hydrazine reduction ($N_2H_4 + 2PDI \rightarrow N_2 + 2$ :$PDI^{2-}/PDI^{2-} + 4H^+$), :$PDI^{2-}$ diradical anions and corresponding $PDI^{2-}$ quinoid tautomer are generated with an emerging sharp absorption profile peaking at 530 nm and 575 nm. Distinct color difference between PDI and :$PDI^{2-}/PDI^{2-}$ aqueous solutions can be distinguished (Supplementary Fig. 2). Electron spin-resonance (ESR) spectra show an intense radical spin feature at $g = 2.0032$ (Fig. 2b), indicating a triplet diradical feature of :$PDI^{2-}$. A number of qualitative spectral characteristics (Supplementary Figs. 2–7) verify the :$PDI^{2-}/PDI^{2-}$ formation and the structure with reduced *cis-/trans-*carbonyls (Fig. 2g) and the simplest protonation form of $(H^+)_2$-(:$PDI^{2-}/PDI^{2-}$). Note that after the two-electron reduction, singlet photoluminescence (PL) emissions at 540 nm and 585 nm corresponding to the radiative relaxation of $S_1 \rightarrow S_O$ can be observed over the :$PDI^{2-}/PDI^{2-}$ solid sample (Fig. 2d), suggesting the coexistence of closed-shell $PDI^{2-}$ quinoid isomers ($S_O$) and open-shell :$PDI^{2-}$ diradicals ($T_1$) through tautomerism. Compared to PDI, :$PDI^{2-}/PDI^{2-}$ exhibits a much smaller Stokes shift (~15 nm) (Supplementary Fig. 8), suggesting a smaller reorganization energy than that of PDI. This is further demonstrated by smaller changes and shifts in infrared vibrational intensity on :$PDI^{2-}/PDI^{2-}$ than PDI under the same excitation condition (Supplementary Fig. 9). Triplet states predominantly decay through non-radiative processes or much slower radiative processes like phosphorescence, resulting in the non-fluorescent :$PDI^{2-}$ diradicals as a 'dark state'. Upon reduction by $N_2H_4 \cdot H_2O$, the fluorescence peak of PDI rapidly decreases to zero within 10 s, resulting in the formation of non-fluorescent :$PDI^{2-}$ triplet diradical anions (Fig. 2e). However, within the next 30 min, fluorescence peaks at 646 nm and 690 nm gradually emerge, indicating the transition from non-fluorescent :$PDI^{2-}$ to fluorescent $PDI^{2-}$ via tautomerism transformation. Such a tautomeric conversion of :$PDI^{2-} \rightarrow PDI^{2-}$ after the 2e⁻ reduction of PDI was further in-situ tracked by the increasing PL emission in 30 min (Fig. 2e). After completely removing residual hydrazine hydrate from the solvent, singlet emissions of the quinoid recovered to 540 nm and 585 nm (Fig. 2f), identical to the solid sample (Fig. 2d). This means it is the co-existed hydrazine hydrate that makes the peaks of $PDI^{2-}$ significant redshift from 540 nm and 585 nm to the 641 nm and 688 nm, as shown in Fig. 2e. To further validate this, we performed additional control experiments by adding varying amounts (0~0.35 µL) of hydrazine hydrate to the DMF solution of the DMF solution of isolated :$PDI^{2-}/PDI^{2-}$ sample. As expected, we observed a gradual redshift in the PL emission of :$PDI^{2-}/PDI^{2-}$ (Supplementary Fig. 10), validating that the presence of hydrazine hydrate in the solution indeed causes a significant redshift in the fluorescence of :$PDI^{2-}/PDI^{2-}$. The diradical/quinoid ratio is determined by the integral PL emission intensity of :$PDI^{2-}/PDI^{2-}$ compared with PDI (:$PDI^{2-}$ is non-fluorescent), suggesting a :$PDI^{2-}/PDI^{2-}$ ratio of 74% to 26% in the precursor samples at room

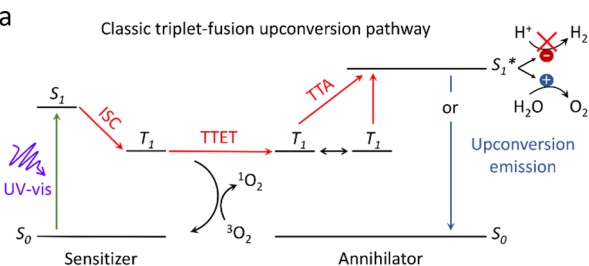

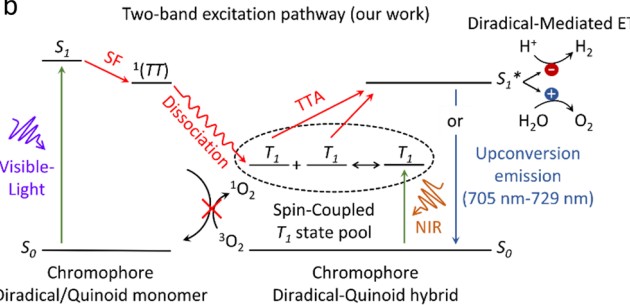

**Fig. 1 | Schematic comparison between different mechanisms. a** The classic triplet-fusion upconversion pathway in an organic sensitizer-annihilator system through ISC, and **b** in this work, the two-band excitation upconversion pathway in a spin-hybrid :$PDI^{2-}/PDI^{2-}$ all-organic semiconductor to utilize the full vis-to-NIR solar spectrum.

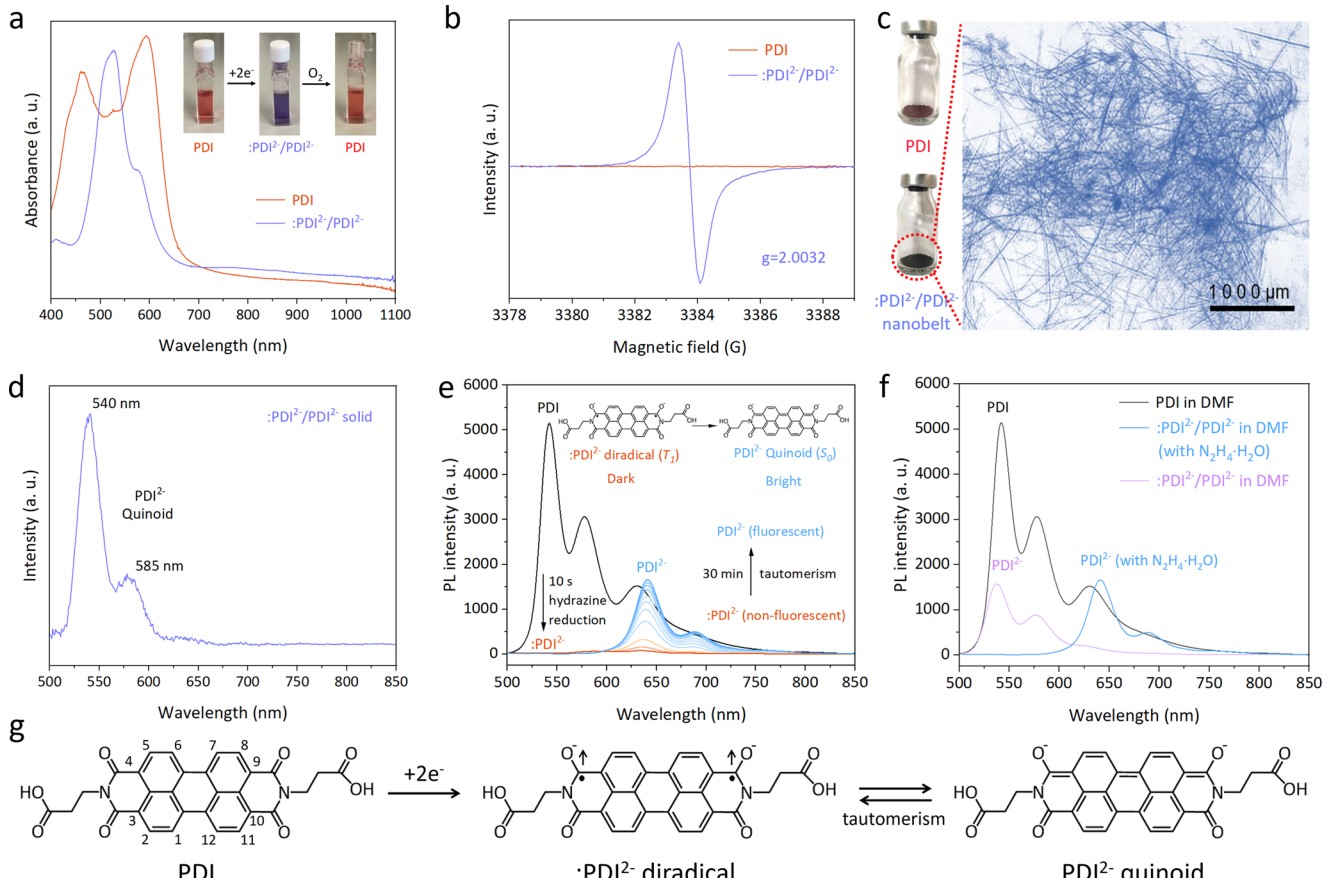

**Fig. 2 | Characterizations of :PDI²⁻/PDI²⁻ precursor. a** UV-vis absorption spectra of PDI and :PDI²⁻/PDI²⁻ precursor in aqueous solutions. Inset shows the color comparison of different solutions. **b** ESR spectra of PDI and :PDI²⁻/PDI²⁻ precursor. **c** Photographs of PDI and self-assembled :PDI²⁻/PDI²⁻ nanobelt (120 h). Inset shows the magnified optical microscope image of the self-assembled :PDI²⁻/PDI²⁻ nanobelt (120 h). **d** The PL spectrum (λ_ex = 450 nm) of :PDI²⁻/PDI²⁻ solid precursor sample. **e** In-situ PL spectra (λ_ex = 450 nm) tracking the emission of PDI after the 2e⁻ reduction by hydrazine hydrate in DMF solution in 30 min. With the addition of 0.25 μL hydrazine hydrate (85%), PDI was first converted to non-fluorescent :PDI²⁻ diradicals. A tautomeric equilibrium from the non-fluorescent diradicals to fluorescent quinoids was then achieved in 30 min. **f** Comparing PL spectra (λ_ex = 450 nm) between pristine PDI and :PDI²⁻/PDI²⁻ samples from in-situ 2e⁻ reduction of PDI for 30 min before and after removing residue hydrazine hydrate in DMF solutions. The hydrazine hydrate component was removed by evaporating the solvent by N₂ flow and then redissolving the sample in an identical concentration. Without hydrazine hydrate, :PDI²⁻/PDI²⁻ sample in DMF shows identical PL emissions to the :PDI²⁻/PDI²⁻ solid sample as in **d**. **g** Schemes show the two-electron reduction pathway of PDI molecules generating :PDI²⁻ diradical anions (T₁) and the tautomeric PDI²⁻ quinoid isomer (S₀).

temperature (15 °C) (Fig. 2f). The solid :PDI²⁻/PDI²⁻ precursor for subsequent self-assembly can be obtained by evaporating the solvent and stabilized under anaerobic conditions.

Due to the solid π-π interaction of perylene rings between individual PDI molecules, the simple phase-transition method has enabled the controllable self-assembly of single-phase PDI-based organic semiconductors with regular morphologies and delicate crystal structures (Supplementary Fig. 11)[26,27]. Applying the same self-assembly strategy, the self-assembly of :PDI²⁻/PDI²⁻ was performed through the phase conversion in an optimized mixture of ethanol (poor solvent) : water (good solvent) = 1 : 1 under anaerobic conditions at 60 °C. Within 120 h, a gradual morphology transition of nanorod→nanowire→nanobelt can be observed with the disappearance of irregular structural fragments (Fig. 3a). Extending the self-assembly time to 120 h renders the formation of single-crystalline nanobelts with a length ≥1 mm (Fig. 2c), the width of ca. -500 nm (Fig. 3a), and the thickness of ca. -250 nm (Fig. 3b). The morphology of self-assembled :PDI²⁻/PDI²⁻ nanobelt no longer changes after 120 h. Separated by centrifugation, obtained all-organic semiconductors remain stable in solvents, including methanol, ethanol, and water, or as a powder for more than half a year without further morphological and structural transitions (Supplementary Fig. 12).

Typically, the self-assembly of PDI derivatives occurs via the strong π-interaction between perylene rings, forming a vertical face-to-face planar stacking *H*-type configuration. In contrast, some polar group-terminated PDI molecules can follow a *J*-type arrangement in a rotational head-to-tail stacking model. N, N'-di(propanoic acid)-perylene-3, 4, 9, 10-tetracarboxylic diimide is well-reported that follows the *H*-type configuration[28]. However, albeit sharing an identical molecular framework with PDI, :PDI²⁻/PDI²⁻ aggregates show a distinct *J*-type pattern (Supplementary Figs. 13, 14 and Supplementary Table 1) with redshifted visible-light absorption after the self-assembly due to π-stacking splitting (see below). Combined with the single-crystal selected area electron diffraction (SAED) pattern (Fig. 3c), a simulated crystal model of :PDI²⁻/PDI²⁻ organic semiconductor can be visualized (Fig. 3d). Notably, after the self-assembly, the fluorescence emissions of the PDI²⁻ quinoid isomer can still be observed (Supplementary Fig. 15), indicating that the tautomeric equilibrium between :PDI²⁻ and PDI²⁻ maintains in self-assembled crystals.

Moreover, after the self-assembly, the lattice parameter of :PDI²⁻/PDI²⁻ crystal evolves as the {100} peak shifts towards larger angles in 120 h (Fig. 3e). Such a shift epitomizes simultaneous features combining the disappearance of irregular fragments and the growth of self-assembled crystals, corresponding to a shortened apparent lattice

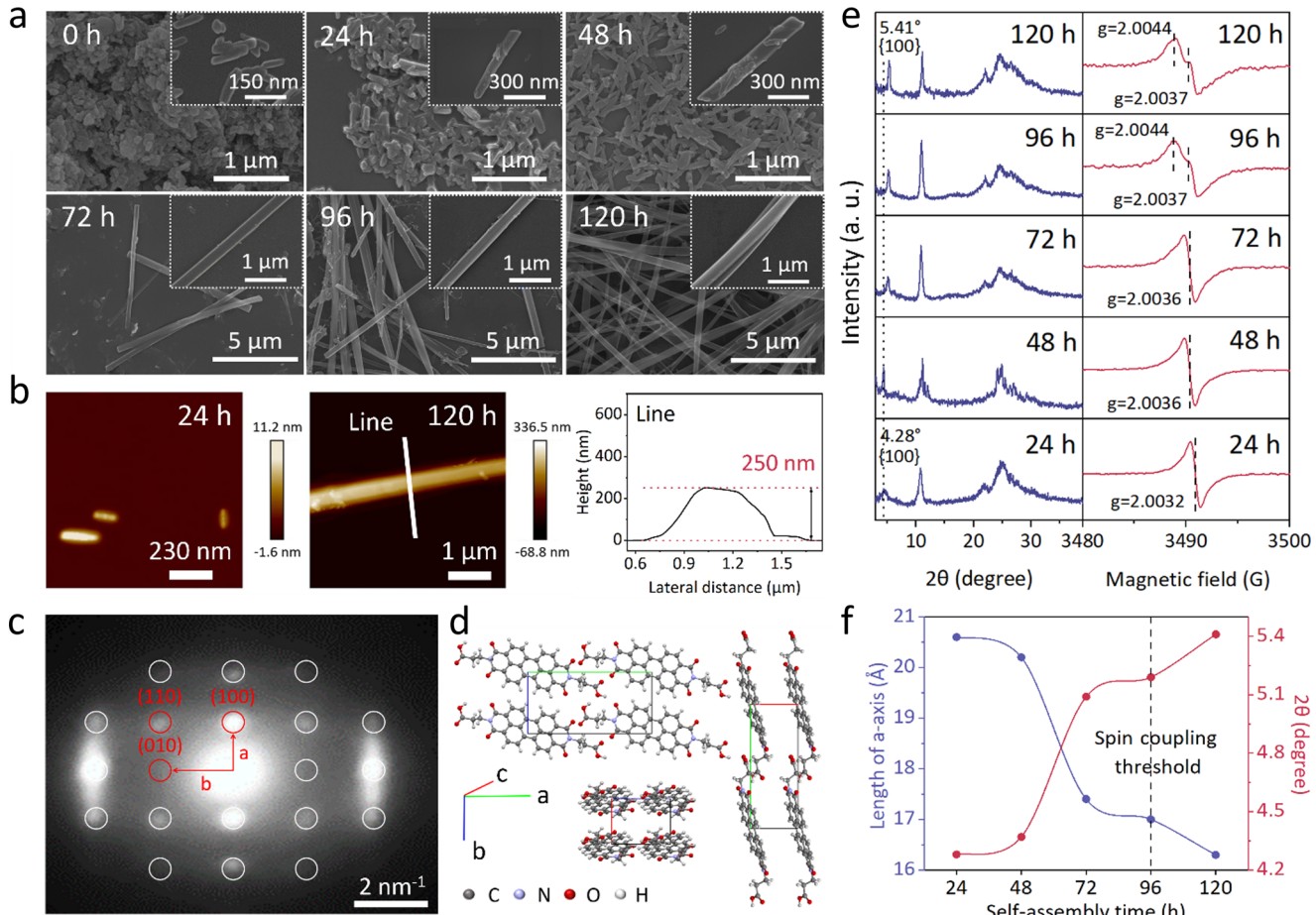

**Fig. 3 | Structural characterizations of self-assembled :PDI²⁻/PDI²⁻ organic semiconductor. a** SEM images to show morphology evolution of ·PDI⁻ during the self-assembly. **b** AFM images of :PDI²⁻/PDI²⁻ samples after 24 h and 120 h self-assembly and corresponding line-scan profile to show the thickness of :PDI²⁻/PDI²⁻ nanobelt. **c** Single-crystal SAED pattern of :PDI²⁻/PDI²⁻ nanobelts (120 h). **d** A simulated arrangement model of :PDI²⁻/PDI²⁻ crystal according to the SAED and XRD patterns. **e** XRD patterns and ESR spectra of :PDI²⁻/PDI²⁻ with different self-assembly times within 120 h. **f** Profiles of the changing a-axis length and 2θ diffraction angle during the self-assembly were calculated according to the {100} peak position in XRD patterns.

length along the a-axis of the self-assembled crystal (Fig. 3f). Meanwhile, the ESR feature of :PDI²⁻ also shifts with the self-assembly time (Fig. 3e). After 96 h of self-assembly, a new splitting feature at $g = 2.0044$ and $g = 2.0037$ corresponding to the non-zero field transition of the spin-triplet state (STS) emerge, which implies a solid spin-orbit coupling within triplet :PDI²⁻ diradicals. Such a triplet spin-orbit coupling can occur on neighboring diradicals or between quinoids and diradicals, forming the triplet-singlet diradical-quinoid (D-Q) hybrid state. Since a shortened a-length from strain narrows the intermolecular distance between building blocks, the interaction between unpaired electrons can be enhanced, leading to an intermolecular triplet spin-orbit coupling within the patterned self-assembled :PDI²⁻/PDI²⁻ crystal. Moreover, a broad NIR absorption band peaking at ~700 nm emerged after the spin coupling, which broadened the visible-light absorption range of self-assembled :PDI²⁻/PDI²⁻ nanobelt close to the profile of the solar spectrum (Fig. 4c). Such an extra NIR absorbance corresponds to an unusual direct $S_O \rightarrow T_n$ transition that is typically spin-forbidden in organic molecular chromophore systems. Due to the forbidden nature of the immediate optical transition between the spin-zero ground ($S_O$) state and the spin-one triplet ($T_1$) levels, molecular triplet exciton is a dark state with no direct triplet absorption[29,30]. However, pioneering works that couple organic excimers with unpaired spins of heavy atoms such as Lanthanide enable a bright triplet NIR absorption through the triplet energy transfer[21–25,29,30]. In our system, the NIR absorption occurs in the

interval of enhanced triplet spin-orbit coupling (72-96 h) (Fig. 5a), consistent with the feature of the direct triplet excitation of $S_O \rightarrow T_n$ transition on the D-Q hybrid. Such a direct $S_O \rightarrow T_1$ transition not only activates 'dark' triplet :PDI²⁻ diradicals to the NIR absorbance, but also provides the opportunity to double the energy of these formed triplet excitons through a TTA upconversion for water-splitting reaction.

**Photocatalytic overall water-splitting**

Figure 4a shows the quantified evolution rates of H₂ and O₂ from 50 mL water containing 0.05 g :PDI²⁻/PDI²⁻ catalysts with different self-assembly times under the constant white light illumination (Xe lamp, central spot intensity 1000 mW cm⁻²), in which H₂ and O₂ productions can be observed without introducing any metal-based cocatalysts. The discrepancy in the H₂/O₂ deviating from the ideal 2 :1 stoichiometric ratio is from the inevitable H₂O₂ production, which accounts for approximately 60 μmol g⁻¹ h⁻¹ (Supplementary Fig. 16 quantified by the I⁻ titration method). The performance of :PDI²⁻/PDI²⁻ catalysts has been improved in order of magnitudes after the self-assembly. Notably, a leap of the performance (~2.5 times) can be observed on :PDI²⁻/PDI²⁻ catalysts between 72 h and 96 h of self-assembly (Fig. 4a), consistent with the interval of triplet spin-orbit coupling (Fig. 3e). Further extending the self-assembly time to 120 h shows the best hydrogen production rate (HPR). As controls, water-splitting experiments on PDI before and after self-assembly were performed with no H₂ and O₂ formation detected (Supplementary Fig. 17). Isotopically labeled

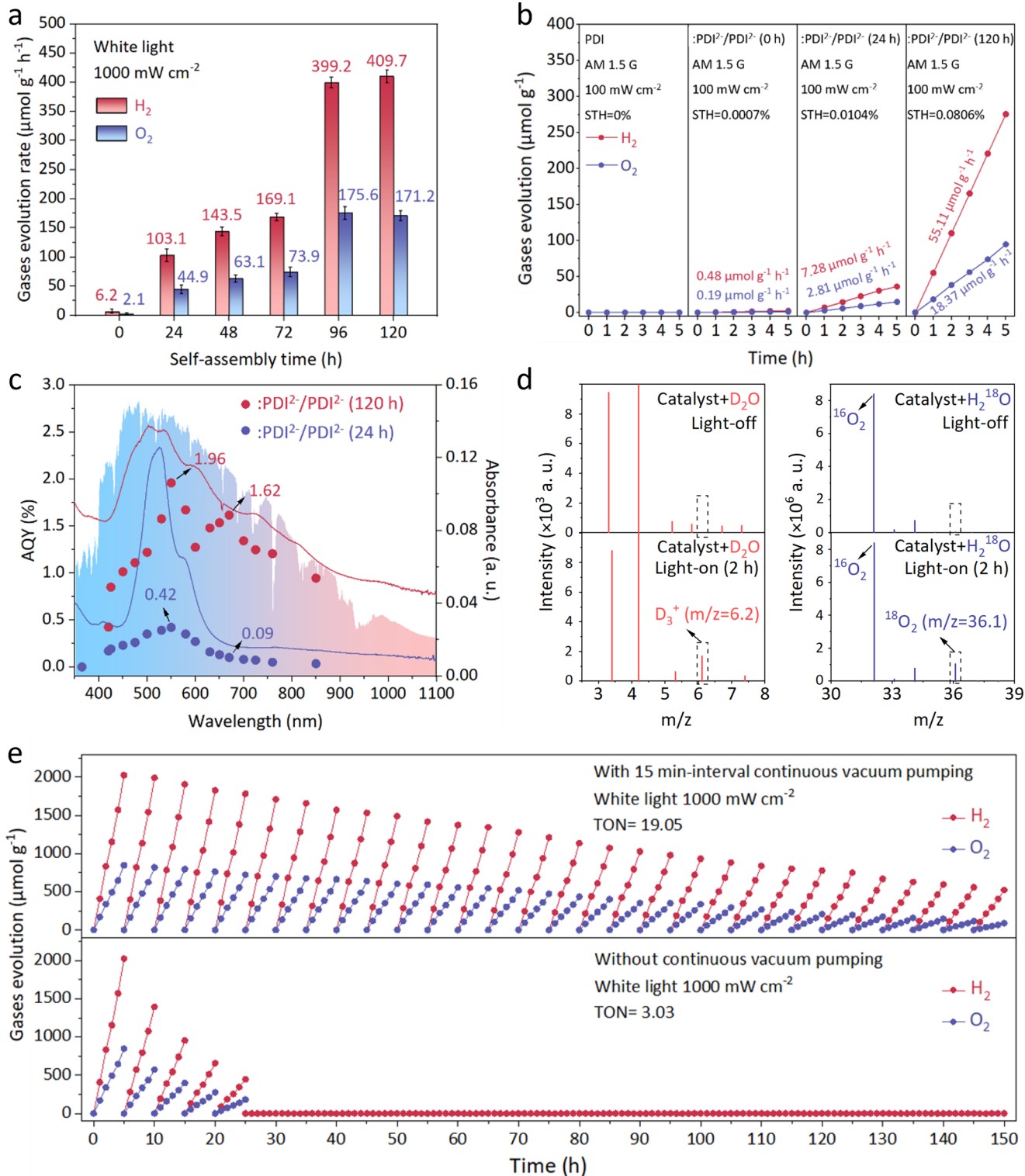

**Fig. 4 | Photocatalytic overall water-splitting over :PDI²⁻/PDI²⁻ organic semi-conductor catalysts. a** Comparisons of H₂ and O₂ productions from pure water on :PDI²⁻/PDI²⁻ catalysts with different self-assembly times (0-120 h) under the white light illumination (≥300 nm, 1000 mW cm⁻²). Error bars (in standard deviation) were obtained by statistically repeating identical experimental results three times. **b** Time-profiles of H₂ and O₂ productions from pure water on PDI, :PDI²⁻/PDI²⁻ without self-assembly, :PDI²⁻/PDI²⁻ nanorod (24 h), and :PDI²⁻/PDI²⁻ nanobelt (120 h) under identical AM 1.5 G simulated solar irradiation (1 sun, 100 mW cm⁻²) (performed once). **c** Wavelength-dependent AQYs for water splitting over :PDI²⁻/PDI²⁻ nanorod (24 h) and :PDI²⁻/PDI²⁻ nanobelt (120 h) along with corresponding UV-vis absorption spectra and the solar spectrum measured at 2 p.m. on June 13, 2022, at Jiangsu University (Zhenjiang, China). **d** GC-MS profiles of isotopically labeled gas products from water-splitting over :PDI²⁻/PDI²⁻ nanobelt (120 h) in deuterium-labeled D₂O and ¹⁸O-labeled H₂¹⁸O at light-off and light-on conditions, respectively. Raw data of the isotopically labeled see Supplementary Figs. 18–22. **e** Long-term recycling gases production performances over :PDI²⁻/PDI²⁻ nanobelts (120 h) under white light (1000 mW cm⁻²) in 150 h with and without 15-min interval continuous vacuum pumping (performed once).

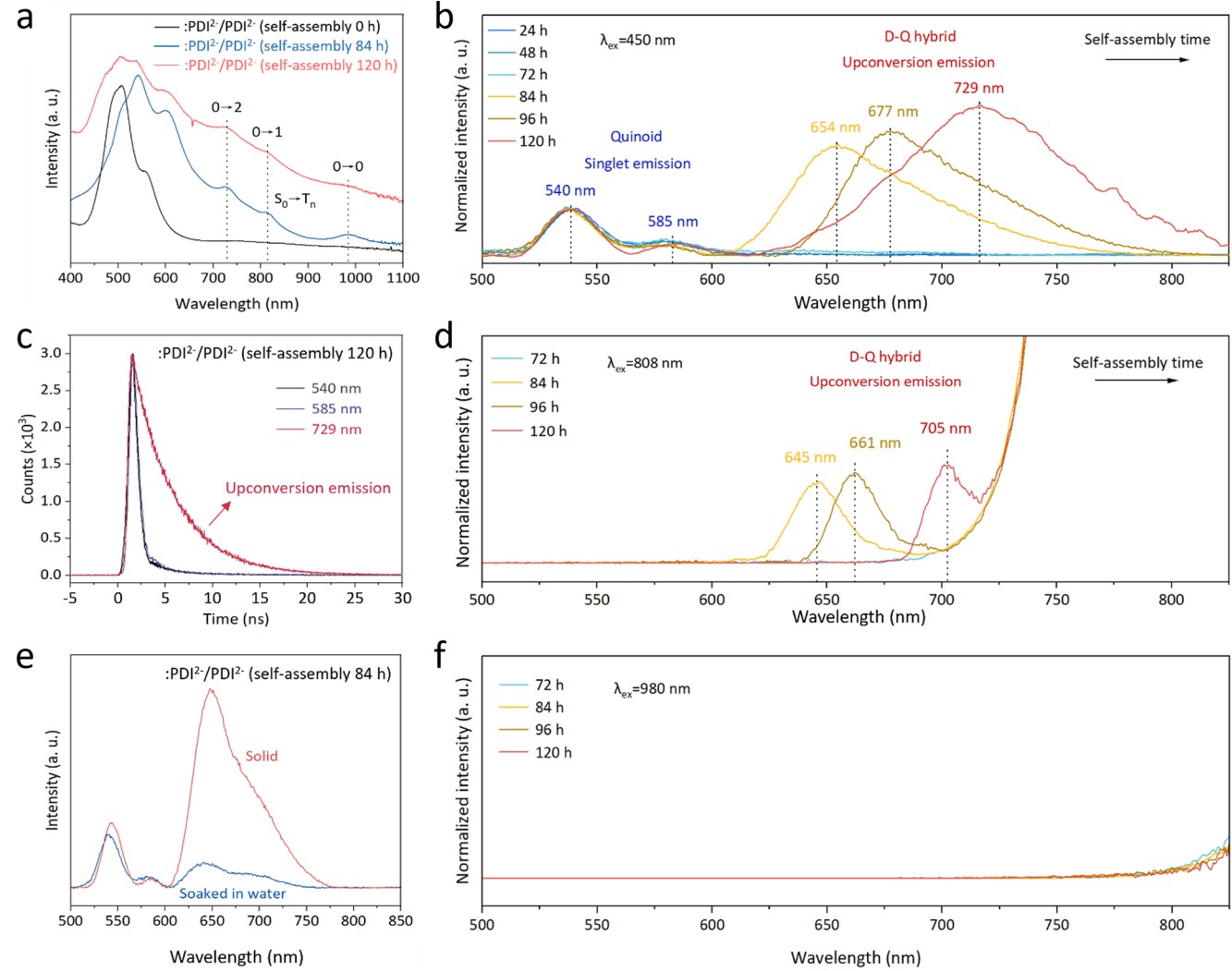

**Fig. 5 | Photoluminescence characterizations of :PDI²⁻/PDI²⁻ organic semiconductors. a** UV-vis absorption spectra of :PDI²⁻/PDI²⁻ with different self-assembly times to show the direct $S_O \rightarrow T_n$ transition. **b** PL spectra ($\lambda_{ex}$ = 450 nm) of :PDI²⁻/PDI²⁻ with different self-assembly times within 120 h. **c** Time-resolved transient PL emission decay ($\lambda_{ex}$ = 450 nm) on :PDI²⁻/PDI²⁻ nanobelt (120 h) with $\lambda_{em}$ = 540 nm, 585 nm, and 729 nm, respectively. **d** Upconversion PL spectra ($\lambda_{ex}$ = 808 nm) of :PDI²⁻/PDI²⁻ with different self-assembly times (72-120 h). **e** PL emission ($\lambda_{ex}$ = 450 nm) of :PDI²⁻/PDI²⁻ (after 84 h of self-assembly) quenched by water. **f** PL spectra ($\lambda_{ex}$ = 980 nm) of :PDI²⁻/PDI²⁻ with different self-assembly times (72-120 h). No upconversion emission was observed.

experiments were further conducted to confirm that observed H₂ and O₂ productions were sorely from the photocatalytic water splitting rather than other effects (Fig. 4d, Supplementary Figs. 18–22). A turnover number (TON) > 19 was achieved within 150 h by performing the long-term reaction over the self-assembled :PDI²⁻/PDI²⁻ nanobelt catalyst, solidly verifying the occurrence of overall water splitting (Fig. 4e). Due to the intense visible-to-NIR absorption in the range of 450 nm–850 nm, self-assembled :PDI²⁻/PDI²⁻ nanobelt catalyst exhibits superior efficiencies under the AM 1.5 G simulated sunlight (1 sun, 100 mW cm⁻²) (Fig. 4b) and apparent quantum yields (AQY) in the visible-to-NIR region (Fig. 4c). The :PDI²⁻/PDI²⁻ nanobelt catalyst exhibits a record AQY of 1.96% at 550 nm and the average AQY exceeds 1.5% in the range of 550–850 nm, the most intense radiation band of the solar spectrum (Supplementary Tables 2–4, performed once). More importantly, with the λ = 850 nm excitation, an AQY of 0.95% was obtained. Such a single-band NIR-driven water-splitting is impossible with direct excitations by low-energy photons (<1.5 eV) due to the overpotential and thermodynamical requirement[1–4]. The STH efficiency of the :PDI²⁻/PDI²⁻ nanobelt catalyst was determined to be ~0.0806%, which is the first example of solar water splitting over all-organic materials. Our self-assembled :PDI²⁻/PDI²⁻ catalyst only stably

works under continuous vacuum pumping to avoid the excessive accumulation of O₂ (Fig. 4e), which can slowly corrode the material.

## Mechanism analysis

We attribute the solar overall water-splitting efficiency over the self-assembled :PDI²⁻/PDI²⁻ catalyst to the following two aspects : i) accommodating the two-band excitation, i.e. spin-allowed SF and spin-forbidden $S_O \rightarrow T_1$ transition, to coherently utilize the full visible-to-NIR solar spectrum to produce triplet excitons for TTA-based upconversion; ii) decreased HER overpotential by hydrophilic diradical :PDI²⁻ moiety with highly delocalized unpaired electrons. We first investigate the excitation feature through the steady-state PL emissions of :PDI²⁻/PDI²⁻ solid samples with different self-assembly times. Under 450 nm excitation (Fig. 5b), :PDI²⁻/PDI²⁻ samples only exhibit singlet emissions of the quinoid moiety at 540 nm and 585 nm with self-assembly times of 24 h, 48 h and 72 h. However, with the increasing self-assembly times, :PDI²⁻/PDI²⁻ samples from 84 h to 120 h additionally show a broad and intense emission band transitioning from 654 nm to 729 nm on the original quinoid singlet emission feature (Fig. 5b), indicating a new $S_1$ state in sharp contrast to that of the quinoid moiety. Such a broad PL emission appears on :PDI²⁻/PDI²⁻ samples after 84 h of self-

assembly, coinciding with the spin coupling threshold (Fig. 3e). Due to the unchanged PL feature of the quinoid moiety and the non-luminescent property of :PDI$^{2-}$ diradicals, we ascribe the newly formed PL emission to the TTA upconversion on the D-Q hybrid (Fig. 5b). Under 450 nm excitation, such a TTA upconversion are only available on well-assembled crystals, which we believe arise from the spin-allowed $S_0S_1 \rightarrow {}^1(TT)$ SF on the diradical chromophore and subsequent ${}^1(TT) \rightarrow T_1 + T_1$ dissociation process to produce the final $T_1$ state (diffusive in the D-Q hybrid). On the 120 h self-assembled :PDI$^{2-}$/PDI$^{2-}$ nanobelt, the quantum efficiency for the TTA upconversion emission ($\lambda_{em}$ = 719 nm) was measured as 0.688%. Typically, two-photon or multi-photon upconversion processes exhibit nonlinear behaviors[31]. However, in our system, the triplet fusion upconversion (TFU) occurs immediately following SF, which differs from the conventional upconversion mechanisms. We further performed PL emission spectra measurements with varying light intensities (10 µJ cm$^{-2}$ ~ 30 µJ cm$^{-2}$) (Supplementary Fig. 23a). The observed linearity does not conform to the typical two-photon upconversion mode, showing a linear relationship (Supplementary Fig. 23b), which is consistent with our SF-TFU transition route. The SF and TFU processes are inversely related and interconnected, leading to a less pronounced nonlinear behavior. Similar phenomenon has been observed on other organic photovoltaic cells that employs a pentacene-based SF process with a sublinear dependence on excitation intensity, which is distinct from the typical nonlinear response in conventional multi-photon upconversion systems[32]. Notably, the lifetime of the upconversion emission remains unchanged with different incident light intensities (Supplementary Fig. 23c), suggesting that no saturation or intensity-induced quenching (such as exciton-exciton annihilation or energy pooling) occurs during this process. When same samples were measured under 808 nm excitation, an identical transitioning PL emission from 645 nm to 705 nm after 84 h of self-assembly was observed with singlet emissions of the quinoid moiety at 540 nm and 585 nm disappeared as expected (Fig. 5d). Since the $S_1$ state of both :PDI$^{2-}$ and PDI$^{2-}$ chromophores are above the 808 nm excitation energy (Fig. 5a), triplet excitons are no longer produced via SF, but from a direct $S_0 \rightarrow T_1$ transition on the D-Q hybrid. This provides solid evidence for the self-assembly-induced triplet-fusion upconversion.

Furthermore, we conducted femtosecond transient absorption spectroscopy (fs-TAS) measurements on the :PDI$^{2-}$/PDI$^{2-}$ samples before and after 120 h self-assembly. The 120 h self-assembled :PDI$^{2-}$/PDI$^{2-}$ sample showed a rapid singlet fission (SF) process (occurring in 300 fs) and the emergence of a triplet excited state absorption (ESA) feature (around 770 nm) after SF upon $\lambda_{ex}$ = 540 nm excitation (Supplementary Fig. 24d–g), with the SF yield calculated to be approximately 42%. This observation contrasts with the behavior of the :PDI$^{2-}$/PDI$^{2-}$ molecular precursor (Supplementary Fig. 24a–c). An identical triplet ESA at ~770 nm, resulting from the direct $S_0$-$T_1$ transition, was also observed under $\lambda_{ex}$ = 700 nm excitation (Supplementary Fig. 24h–j). Detailed data analysis and signal assignment was described in Supplementary Information. Our fs-TAS observations are consistent with our photoluminescence (PL) emission experiments and strongly support our proposed mechanism.

The transitioning redshift of the TTA-upconversion emission is due to the enhancing spin coupling between diradicals and quinoids with the increasing self-assembly times, similar to that in the rubrene-lanthanide system[21]. Furthermore, the upconversion emission of the 120 h self-assembled sample at 729 nm has a much longer life-time ($\tau$ = 4.33 ns) than that at 540 nm ($\tau$ = 0.67 ns) and 585 nm ($\tau$ = 0.79 ns) (Fig. 5c). A longer emission decay time directly corresponds to a slower back-electron-transfer rate[33], which facilitates the ns-scale interfacial water-splitting reaction. This new $S_1$ state on the D-Q hybrid with a maximum 729 nm emission indicates that the energy of the parent $T_1$ state is lower than 0.85 eV, which is below the ${}^3O_2 \rightarrow {}^1O_2$ (0.94 eV) quenching energy and guarantees no interference of formed

$O_2$[34]. More importantly, the observed TTA upconversion emission could be severely extinguished by water (Fig. 5e), while singlet emissions at 540 nm and 585 nm of quinoid moiety could not. This indicates that the new $S_1$ excited state on the D-Q hybrid from TTA-upconversion is water-reactive, consistent with the unusual NIR-driven water-splitting efficiency on :PDI$^{2-}$/PDI$^{2-}$ nanobelt. We further conducted temperature-dependent PL measurements on the :PDI$^{2-}$/PDI$^{2-}$ (120 h) sample over 100 K to 300 K (Supplementary Fig. 25a). Through the Arrhenius kinetic analysis, the binding energy ($E_b$) of *$S_1$ singlet excitons after the TTA upconversion is determined as $21 \pm 2$ meV (Supplementary Fig. 25b), which is significantly lower than the binding energies typically observed in other organic semiconductors. This value is less than the $E_b$ of the $S_1$ state associated with the 540/580 nm emission and the $K_BT$ value (~26 meV) at room temperature, suggesting that TTA-resulted *$S_1$ singlet excitons are highly active and readily dissociate into electrons and holes to participate in reactions, which is consistent with the fluorescence quenching in water (Fig. 5e). The self-assembly of such spin-coupled :PDI$^{2-}$/PDI$^{2-}$ chromophore renders intense adsorption features in the NIR region (700 nm-1100 nm) (Fig. 5a), matching well with the direct $S_0 \rightarrow T_n$ transition feature with 0-0, 0-1, and 0-2 vibrational states[21–25]. However, with the 980 nm excitation (~lowest 0-0 vibrational level), the upconversion emission on samples did not appear (Fig. 5f), implying that the TTA-upconversion requires photon energies sufficient for the $S_0 \rightarrow T_1$ transition.

Besides the two-band excitation to make the full use of the visible-to-NIR solar spectrum, breaking through the HER bottleneck with even a much lower $S_1$ state level on the D-Q hybrid than that either on :PDI$^{2-}$ or PDI$^{2-}$ monomer chromophore is another important issue. A radical-mediated electron transfer (ET) pathway is reported to hold much lower reaction overpotentials, allowing reactions that are impossible under the non-radical pathway to occur[35,36]. In our system, the interfacial ET of HER occurs on the ordered :PDI$^{2-}$/PDI$^{2-}$ surface with plenty of diradical unpaired electrons, which makes the H$^+ \rightarrow$ H$_2$ reduction easier along the favorable diradical-mediated pathway. In addition, the enhanced hydrophilicity of :PDI$^{2-}$/PDI$^{2-}$ nanobelt may also play an important role. Unlike the generally inert covalent H-coordination of organic surface, :PDI$^{2-}$/PDI$^{2-}$ nanobelt has an ionic diradical dipole with enhanced affinity with protons to reach the charge balance, forming a hydrophilic surface with better wettability (Supplementary Fig. 26), which is essential for the adsorption and dissociation of H$_2$O molecules at the water/catalyst interface. Moreover, the small $E_b$ of TFU-resulted *$S_1$ excitons ($21 \pm 2$ meV) makes the reduced :PDI$^{2-}$ diradicals exhibit properties more akin to inorganic materials, contributing to the breaking through of HER bottleneck without noble-/transition-metal components.

Our work is the proof-of-concept example of a metal-free organic catalyst capable of vis-to-NIR-driven overall water-splitting. Furthermore, such a self-assembly strategy of open-shell/closed-shell resonant structure applies to multiple derivatives of the PDI family (five derivatives verified) (Supplementary Fig. 27). This provides a path to design a series of brand-new photo-response organic semiconductors for photocatalysis, solar cells, sensors and light emitting materials.

## Methods

### Chemicals

Perylene-3, 4, 9, 10-tetracarboxylic dianhydride (≥92.0%), β-Alanine (99%), hydrochloric acid (HCl, 37%), hydrazine hydrate (N$_2$H$_4$·H$_2$O, >85%), imidazole (99%), 1, 10-phenanthroline (99%), Fe(NO$_3$)$_3$ (99.99%), sodium acetate (99%), tetrabutylammonium tetra-fluoroborate (≥98%), dimethyl sulfoxide (AR, >99%) and acetic acid (GR, 99.8%) were all purchased from Aladdin Biochemical Technology Co., Ltd. China. Deuterium oxide (D$_2$O, 99 atom % D) and heavy oxygen water (H$_2$$^{18}$O, 97 atom % $^{18}$O) were obtained from Sigma-Aldrich Company Ltd. China.

## Preparation of PDI

In a typical preparation procedure, 1.376 g (3.507 mmol) of perylene-3,4,9,10-tetracarboxylic dianhydride, 2.5 g (28.06 mmol) of β-Alanine, 18 g of imidazole were placed in a mortar and evenly grounded. Immediately transferred to a porcelain boat, heated in a tube furnace at 100 °C for four hours under the protection of Nitrogen. The resulting solid product was then uniformly dispersed in 100 mL of ethanol with the addition of 300 mL of 2 M HCl. After magnetically stirring overnight, the resulting mixture was centrifuged and washed with deionized water until the solution was neutral (pH ~7). Obtained solid was collected and dried at 60 °C in the vacuum for further use.

## Preparation of :PDI$^{2-}$/PDI$^{2-}$ precursor

In a typical preparation procedure, 1.0 g of freshly prepared PDI was first dissolved uniformly in 100 mL hydrazine hydrate (>85%). The reduction reaction should last for more than 4 h. Then evaporating the solvent with the protection of Nitrogen flow. Obtained solid was collected and washed with ethanol, and dried on a 60 °C hot stage until constant weight (in a Nitrogen glove box). The obtained product is a mixture of open-shell diradical and the tautomeric closed-shell quinoid isomer with a ratio of 74% to 26%, denoted :PDI$^{2-}$/PDI$^{2-}$, which can be stably stored under the anaerobic condition.

## Self-assembly of :PDI$^{2-}$/PDI$^{2-}$ nanobelt

In a typical preparation procedure, 10 mg :PDI$^{2-}$/PDI$^{2-}$ precursor was dissolved uniformly in 5 mL of deionized water (good solvent), then 5 mL ethanol (poor solvent) was added. The mixed solution was transferred into a 10 mL vial. The vials were sealed and placed on a 60 °C hot stage for 0–120 h. The samples with different self-assembly times were centrifuged and washed with water and dried until constant weight. The above operations were carried out in the Nitrogen glove box.

## Water-splitting performance test

The photocatalytic water-splitting over different catalysts was carried out in an outer irradiation-type photoreactor (quartz glass) connected to a closed gas-circulation system. In a typical procedure, 0.05 g of the photocatalysts were dispersed by a magnetic stirrer in 50 mL of deionized water. The catalyst was transferred from the glove box by injection sampling to avoid oxidation of the catalyst. The suspension is thoroughly degassed to remove air and irradiated using a 300 W Xe-lamp (1000 mW cm$^{-2}$, CEL-HXF300-T3, Beijing China Education Au-light Technology Co. Ltd, China). The photocatalytic H$_2$ and O$_2$ evolution rate was analyzed using an online GC-7920 gas chromatograph (GC, TCD detector, 5 A° molecular sieve columns and Ar$_2$ carrier). Magnetic stirring (400 rpm) was used during the water splitting experiment to ensure homogeneity of the suspension and to eliminate sedimentation. With simulated AM1.5 G sunlight irradiation, a solar simulator (100 mW cm$^{-2}$, Xenon Light Source CEl-S500-T5, Beijing China Education Au-light Technology Co. Ltd, China) was employed as the light source.

## Isotopically labeled GC-MS measurement

**Preparation of isotopically labeled samples.** In the Nitrogen glove box, 20 mg :PDI$^{2-}$/PDI$^{2-}$ nanobelts was dispersed uniformly in 10 mL of D$_2$O (H$_2$$^{18}$O or H$_2$O) and transferred into a 50 mL volume vial. The gas (5 mL) was collected by the injection needle with a throttle valve into sealed gas bag, denoted as Light-off sample. The vials were sealed and irradiated using a 300 W Xe-lamp (1000 mW cm$^{-2}$, CEL-HXF300-T3, Beijing China Education Au-light Technology Co. Ltd, China) for 2 h. The reaction gas (5 mL) was collected by the injection needle with a throttle valve into sealed gas bag, denoted as Light-on sample. The Light-off samples and Light-on samples were collected from the same vial.

## Using agilent CP-Molsieve 5A as a column for isotope-labeled samples analysis

The 2 mL gas samples were collected and manually injected by gas-tight syringes (the FTFE luer lok (PN5190-1534), 0-2.5 mL) and then analyzed by gas chromatography-mass spectrometry (8890-5977B GC-MS instrument, Agilent Technologies, USA) equipped with an Agilent CP-Molsieve 5A column (25 m × 250 μm × 30 μm, Agilent Technologies, USA) in GC-MS. Helium was used as carrier gas. The column was maintained at 100 °C for 5 min, and the flow of the carrier was 1.5 mL min$^{-1}$. The temperatures of the injector and MS source were set to be 100 °C and 230 °C, respectively. The ionization potentials used for analysis were set to 20 eV and 70 eV, respectively.

In addition, we repeated the isotopically labeled GC-MS measurement using headspace vial (sample through drainage method) under above mentioned chromatographic conditions. The ionization potentials used for analysis were set to 18 eV and 70 eV, respectively.

## Fe(III)-1, 10-Phenanthroline spectrometric titration measurements

The reductive electron concentration on :PDI$^{2-}$/PDI$^{2-}$ precursor was measured by a Fe(III)-1, 10-phenanthroline titration spectrometric method. 1, 10-phenanthroline spectrometric measurement is a simple and widely used method for measuring Fe(II) ions. Here, we use the Fe(III) solution to titrate the unpaired electrons on :PDI$^{2-}$/PDI$^{2-}$ precursor that quantitatively lead to the production of Fe(II) ion, then used 1, 10-phenanthroline to measure the concentration of Fe(II) ions. Thus, we can quantitatively obtain the concentration of trapped electrons. The concentration of Fe(NO$_3$)$_3$ solutions employed in this measurement is 10$^{-3}$ M. 0.2% 1, 10-phenanthroline aqueous solution and pH = 4.6 HAc-NaAc buffer solution were previously prepared for use. The pH=4.6 HAc-NaAc buffer solution was prepared by dissolving 135 g sodium acetate and 120 mL acetic acid into 500 mL water solution. Before the titration, the Fe(NO$_3$)$_3$ solution is purged by nitrogen for 30 min to remove oxygen. The titration was conducted in the glove box. In a typical procedure, 2.5 mL :PDI$^{2-}$/PDI$^{2-}$ aqueous solution was taken in the glove box and mixed with 2.5 mL Fe(NO$_3$)$_3$ solution. 5 min later, the resulting mixed solution was taken out to conduct the spectrometric measurement. The spectrometric measurement was conducted in the air. 1.5 mL supernatant taken from the glove box was mixed with 1.5 mL pH = 4.6 HAc-NaAc buffer solution, and then added with 1 mL 0.2% 1, 10-phenanthroline water solution to obtain a red solution. Waiting 5 min to obtain a stable state, the resulting solution was transferred in a quartz cuvette and measured on a Hitachi U3900 spectroscopy. Obtained absorbance value was compared with the standard fitting line to obtain a certain Fe(II) concentration. The background data was collected by the same method except replacing 1, 10-phenanthroline solution with deionized water to eliminate the influence of PDI absorbance.

## Characterization devices

The crystalline structure of the resultant products was characterized by X-ray diffraction (XRD) using an X'Pert-ProMPD (Holand) D/max-γA X-ray diffractometer with Cu Kα radiation (λ = 0.154178 nm) at a scan rate of 5° min$^{-1}$. The near-ambient pressure x-ray photoelectron spectroscopy (NAP-XPS) experiments were conducted on a laboratory-based SPECS near-ambient pressure XPS system. X-ray photoelectron spectroscopy (XPS) measurements were performed using a Nexsa instrument (Thermo Fisher Scientific) with Al Kα radiation (hυ = 1486.6 eV) as the excitation source. Scanning electron microscopy (SEM) images were performed by a FEI-quanta 200 scanning electron microscope with an acceleration voltage of 20 kV. The electron paramagnetic resonance (EPR) analysis was performed on a Bruker EPR JES-FA200 spectrometer. Transient fluorescence decay spectra were characterized on an Edinburgh FLS1000 fluorescence spectrometer. The ultraviolet-visible diffuse reflectance spectroscopy (UV-vis DRS)

was recorded on a Shimadzu UV-3600 spectrometer. Atomic force microscopy (AFM) was performed by means of a Veeco DI Nanoscope Multi Mode V system. The isotopically labeled experiments were carried out on a gas chromatography-mass spectrometry (GC-MS, Agilent 7890B-5977B).

## AQY calculations

The AQYs at different wavelengths ($\lambda = 365$–850 nm) for $:PDI^{2-}/PDI^{2-}$ catalysts were determined by taking catalyst solution (0.05 g catalyst and 50 mL deionized water) irradiated by a 300 W Xe lamp applying different band-pass filters for 5 h. The average intensity of irradiation was determined by a CEL-NP2000-10A spectroradiometer (Beijing China Education Au·light Technology Co. Ltd, China) and the irradiation area was 4 cm² (2 cm × 2 cm). For example, when $\lambda = 550$ nm, the average intensity of irradiation was determined to be 269.8 mW cm⁻². The amount of $H_2$ generated in 5 h was 87.5 μmol. The number of incident photons (N) is as calculated by the following equation:

$$N = \frac{E\lambda}{hc} = \frac{269.8 \times 4 \times 10^{-3} \times 5 \times 3600 \times 550 \times 10^{-9}}{6.626 \times 10^{-34} \times 3 \times 10^{8}} = 0.5375 \times 10^{22} \quad (1)$$

So,

$$
\begin{aligned}
AQY &= \frac{2 \times \text{the number of evolved } H_2 \text{ molecules}}{N} \times 100\% \\
&= \frac{2 \times 6.02 \times 10^{23} \times 87.5 \times 10^{-6}}{0.5375 \times 10^{22}} \times 100\% = 1.96\%
\end{aligned}
\quad (2)
$$

## STH calculations

The STH efficiency was evaluated using AM 1.5 G solar simulator (1 sun) as the light source with an irradiation area of 2.25 cm² (1.5 cm × 1.5 cm). With $:PDI^{2-}/PDI^{2-}$ nanobelt (120 h) as the catalyst (0.05 g catalyst in 50 mL water), the $H_2$ generated in 5 h was 275.55 μmol ($R_{H2} = 7.654 \times 10^{-7}$ mmol s⁻¹).

So,

$$
\begin{aligned}
STH &= \frac{\Delta G_r \times R_{H2}}{P_{Sun} \times S} \times 100\% = \frac{2.37 \times 10^3 \times R_{H2}}{S} \times 100\% \\
STH &= \frac{2.37 \times 10^3 \times 7.654 \times 10^{-7}}{2.25} \times 100\% = 0.0806\%
\end{aligned}
\quad (3)
$$

Here, $R_{H_2}$: The $H_2$ evolution rate (mmol s⁻¹);

$\Delta G_r$: The reaction Gibbs energy of water splitting (J mol⁻¹);

$P_{Sun}$: Light power density of the AM 1.5 G standard solar spectrum (100 mW cm⁻²);

S: The irradiation area.

# Data availability

The data supporting the findings of this study are available within the article and its Supplementary Information files. All other relevant source data are available from the corresponding author upon request. Source data file has been uploaded. Source data are provided with this paper.

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

## Acknowledgements

We gratefully acknowledge the financial support of the Strategic Priority Research Program of the Chinese Academy of Sciences (Grant No. XDB3600000 (J.Z.)), National Natural Science Foundation of China (Grant No. 21776117 (P.H.), 21806060 (Y.Y.), 22176192 (W.M.), 22076007 (D.M.)). Advanced Talent Foundation of Jiangsu University (Grant. No. 22JDG017 (Y.Y.)). We gratefully appreciate the guidance of Dr. Dujuan Yang for GC-MS measurement and Dr. Fengfeng Yang for GC-MS measurement using headspace vial at the Agilent Shared Laboratory (ASL) in Shanghai. We are grateful for the technical support for Nano-X from Suzhou Institute of Nano-Tech and Nano-Bionics, Chinese Academy of Sciences (SINANO).

## Author contributions

X. Lin designed the photocatalyst and performed the photocatalytic reactions. Y. Hao and Y. Gong designed and synthesized the PDI. P. Zhou performed crystal structure simulation. D. Ma and P. Huo performed AQY and STH measurements. X. Lin and Y. Hao performed PL measurements. Z. Liu, Y. Sun and H. Sun performed SEM, XRD and ESR. Y. Chen and S. Jia performed photocatalytic experiments. W. Li, C. Guo and Y. Zhou synthesized the PDI1-PDI5. S. Yuan and J. Zhao supervised the whole project. X. Lin, Y. Yan and W. Ma wrote the manuscript with input from all authors.

## Competing interests

The authors declare no competing interests.
