## [Peer Review File · Nature Communications]

REVIEWER COMMENTS

Reviewer #1 (Remarks to the Author):

Compared to the conventional inorganic photocatalysts like TiO₂, organic semiconductor materials offer some superior features for application in photocatalysis, particularly splitting of water to hydrogen. For example, organic molecules are more tunable for electronic structure allowing for broader light absorption. Moreover, most of organic semiconductors are based on pi-conjugate structure that is conducive to creation and stabilization of radical state through large area pi-electron delocalization. Taking this unique feature, the authors of this paper report on an innovative approach to co-assembly of diradical and quinoid anions of a perylene diimide molecule into shape-defined nanobelts. The tunable spin states provide effective photocatalysis activity in vis-NIR region as evidenced for the STH process. Such a proof-of-concept success not only paves a new way for design and development of organic photocatalysts, but also helps open a novel scenario for improving the photocatalysis efficiency by tuning the spin states.

Reviewer #2 (Remarks to the Author):

Xinyu Lin et al. report on the sunlight-driven photocatalytic water-splitting using an organic semiconductor photocatalysts consisting of a mixture of the open-shell diradical and the tautomeric closed-shell quinoid isomer forms of the reduced conjugated small molecule perylene-3, 4, 9, 10-tetracarboxylic diimide (PDI).

There have been prior reports of water-splitting using perylene-based organic semiconductors. What is somewhat new here is the explicit use of singlet fission and triplet fusion processes to achieve the charge carriers capable of driving the water-splitting reaction.

Unfortunately, the manuscript is written in a convoluted manner which makes it hard to follow the central hypothesis and the evidence collected to substantiate it.

The noteworthy result is the demonstration of a fluorescent modification of the well-known PDI nanobelts through hydrazine hydrate reduction.

I believe that publication is premature because of the following glaring deficiencies in the manuscript:

1. Organic semiconductors have large exciton binding energies which range from 0.2 eV to 1.2 eV. Even assuming that the triplet fusion process is efficient in generating S₁* singlet excitons, there is no mention of the binding energy of these excitons or the mechanism by which the bound electron-hole pair in the Frenkel-type excitons is separated. There is absolutely no characterization data that sheds light on this critical step.
2. The authors state that the "molecular triplet exciton is a dark state with no direct triplet absorption", which is not entirely true. The triplet state certainly has an extremely weak oscillator strength but is typically accessible through high intensity laser absorption or ultrafast pump-probe transient absorption spectroscopy (TAS). It is somewhat surprising that the authors have not collected & analyzed fs TAS data

in this work.

3. The maximum quantum yield reported by the authors (1.5%) is extremely low and very far from practical use. The weak water-splitting performance also has consequences for the science described in this work, in the sense that a low quantum yield water-splitting could be generated through a regular singlet exciton generation and dissociation process without the need to invoke singlet fission or triplet fusion.

4. H₂ and O₂ are claimed to be generated in a perfect 2:1 stoichiometric ratio but Figure 3a shows otherwise. In fact, the ratio varies from 2.3 to 3. Why?

5. LACK OF QUANTIFICATION: The yields of the intersystem crossing process, singlet fission process and triplet fusion process are not provided.

OTHER SCIENTIFIC QUESTIONS

6. In Figure 1a and 1d, the absorption and PL spectra of :PDI2-/PDI2- look too similar, suggesting a negligible Stokes shift, which would be unusual in organic semiconductors that tend to exhibit a large reorganization energy. This aspect requires more comment and elaboration. It is highly recommended that the authors collect Raman spectra from the sample, since this data provided an independent method to obtain the reorganization energy.

7. The authors claim that :PDI2-/PDI2 is soluble in water. Photographs and spectra of the aqueous solutions should be provided, at least in Supporting Information.

8. Upconversion processes are known to be nonlinear. Therefore it is really important that the authors measure & report the emission quantum yields & lifetimes of the upconversion luminescence as a function of excitation intensity over at least 3 orders of magnitude of intensity. In other words, I am asking for Figures 4c -to 4f to improved and augmented.

Reviewer #3 (Remarks to the Author):

Lin et al. reported a direct solar-to-hydrogen conversion from pure water using an all-organic heterogeneous catalyst. Although the results sound interesting, the current study is incomplete and has many flaws and errors. Therefore, the current version cannot be published in Nature Communications, and authors should be encouraged to do more work to prove whether the work is suitable for publication in the future. The details are as follows:

The authors obtained PDI2- by introducing a strong alkaline reducing agent, and PDI2- is not stable in air, which greatly weakens the highlight of the article and reduces the real application of the material in water-splitting reaction. The authors should consider how to make PDI2- relatively stable in the air or in-situ formation rather than by chemical reduction.

In Figure 1e, PDI could be reduced by N₂H₄ within 30 min; the fluorescent peak at 650 nm could be attributed to the PDI2-. This is undoubtedly true because a similar phenomenon has been found in the reported article (Angew. Chem. Int. Ed. 2022, 61, e202110491). However, once the PDI2- was transferred to the DMF, the fluorescent peak at 650 nm showed a blueshift to 540 nm, just like the PDI (Figure 1f). The authors still attribute this peak to PDI2-. This is a completely wrong conclusion; PDI has been

oxidized at this time, and the author needs to reconsider this conclusion.

During self-assembly, the authors believe that PDI²⁻ is assembled and the rod-like material is the PDI²⁻. But according to our experiments, the same assembly can occur with PDI. Electron microscopy cannot distinguish between PDI²⁻ and PDI. In addition, PDI is an acid, and its assembly structure is destroyed while being reduced by an alkali-reducing agent, so a primitive disorder occurs. When placed in an aqueous solution of alcohol, it is reassembled, which is not a novelty and is not unique to PDI²⁻.

The isotope tracing experiments the authors carried out are also not credible. Firstly, the authors describe that the details of the isotope labeling experiment are extremely unclear, and there is no way to obtain reliable data for such a crude experiment. Secondly, Figure 3d (left) mentions the MS could be obtained from GC-MS. The carrier gas of GC-MS is He, and the mass-to-charge ratio of He is 4 and under the action of an ion source (ionization process), deuterium can react with residual oxygen in the mass spectrum (there is no absolute vacuum on Earth) to produce D₂O. So, it doesn't make sense to provide such a mass spectrum. Thirdly, Figure 3d (right) the authors mention using the injection method to inject samples. How did the authors prevent oxygen from the air from entering GC-MS? Based on our experiments, neither airtight needles (oxygen in the tip of the needle) nor headspace injection can avoid oxygen interference in the experiment. The author must give a reasonable explanation for these unreasonable results.

Reply to reviewers' comments

To Reviewer 1:

Comments:

Compared to the conventional inorganic photocatalysts like TiO₂, organic semiconductor materials offer some superior features for application in photocatalysis, particularly splitting of water to hydrogen. For example, organic molecules are more tunable for electronic structure allowing for broader light absorption. Moreover, most of organic semiconductors are based on pi-conjugate structure that is conducive to creation and stabilization of radical state through large area pi-electron delocalization. Taking this unique feature, the authors of this paper report on an innovative approach to co-assembly of diradical and quinoid anions of a perylene diimide molecule into shape-defined nanobelts. The tunable spin states provide effective photocatalysis activity in vis-NIR region as evidenced for the STH process. Such a proof-of-concept success not only paves a new way for design and development of organic photocatalysts, but also helps open a novel scenario for improving the photocatalysis efficiency by tuning the spin states

Response: We appreciate the reviewer's positive comments.

To Reviewer 2:

Comments:

Xinyu Lin et al. report on the sunlight-driven photocatalytic water-splitting using an organic semiconductor photocatalysts consisting of a mixture of the open-shell diradical and the tautomeric closed-shell quinoid isomer forms of the reduced conjugated small molecule perylene-3, 4, 9, 10-tetracarboxylic diimide (PDI). There have been prior reports of water-splitting using perylene-based organic semiconductors. What is somewhat new here is the explicit use of singlet fission and triplet fusion processes to achieve the charge carriers capable of driving the water-splitting reaction. Unfortunately, the manuscript is written in a convoluted manner which makes it hard to follow the central hypothesis and the evidence collected to substantiate it. The noteworthy result is the demonstration of a fluorescent modification of the well-known PDI nanobelts through hydrazine hydrate reduction.

I believe that publication is premature because of the following glaring deficiencies in the manuscript:

Q1: *Organic semiconductors have large exciton binding energies which range from 0.2 eV to 1.2 eV. Even assuming that the triplet fusion process is efficient in generating S₁* singlet excitons, there is no mention of the binding energy of these excitons or the mechanism by which the bound electron-hole pair in the Frenkel-type excitons is separated. There is absolutely no characterization data that sheds light on this critical step.*

Response: We appreciate the reviewer's comment regarding the binding energy of *S_1 singlet excitons and the separation mechanism in our self-assembled spin-hybrid: PDI^{2-}/PDI^{2-} crystal nanobelt. You are correct that the binding energy (E_b) of *S_1 singlet excitons after the triplet fusion upconversion (TFU) process is crucial for understanding TFU-driven reactions in organic semiconductors like PDI.

Typically, the singlet E_b of organic semiconductors can be determined by comparing the energy gap between their absorption and PL spectra. Alternatively, for a TFU singlet excited state, the E_b can be estimated through the Arrhenius kinetic analysis of temperature-dependent PL spectra. We carried out PL measurements on the $:PDI^{2-}/PDI^{2-}$ (120 h) sample over a temperature range of 100 K to 300 K (**Figure R1a**). The obtained data were fitted using the equation $\ln(k) = \ln(A) - E_b/RT$, which allowed us to calculate the E_b of TFU-resulted *S_1 singlet excitons as 21 ± 2 meV (**Figure R1b**), which is significantly lower than the binding energies typically observed in other organic semiconductors.

Notably, this value is less than the $K_B T$ value (~ 26 meV) at room temperature, suggesting that TFU-resulted *S_1 singlet excitons are highly active and readily dissociate into electrons and holes to participate in reactions. Moreover, the E_b of the S_1 state associated with the 540/580 nm fluorescence is higher than that of the TFU-resulted *S_1 singlet excitons and the $K_B T$ value (**Figure R1b**), which is consistent with the experimental results of fluorescence quenching in water (**Figure 4e**), where the 719 nm fluorescence is more reactive towards water. This reveals why our all-organic semiconductor material can achieve a breakthrough in the HER bottleneck and overall water splitting without the need for metals. The reduced $:PDI^{2-}$ diradicals exhibit properties more akin to inorganic materials, highlighting the uniqueness and innovation of our spin-coupled organic semiconductor catalyst material. For detailed data analysis and descriptions, please refer to Page 6, Line 17~19, Page 13, Line 28~32, Page 14, Line 1~3 and Page 14, Line 18-20 in the revised manuscript.

Figure R1. (a) Temperature-dependent PL emission spectra ($\lambda_{ex} = 450$ nm) of $:PDI^{2-}/PDI^{2-}$ catalyst (120 h) in the temperature range of 100 K \sim 300 K; (b) Integrated PL intensity from (a) as a function of temperature. E_b is calculated by Arrhenius plot fittings.

Q2: *The authors state that the "molecular triplet exciton is a dark state with no direct triplet absorption", which is not entirely true. The triplet state certainly has an extremely weak oscillator strength but is typically accessible through high intensity laser absorption or ultrafast pump-probe transient absorption spectroscopy (TAS). It is somewhat surprising that the authors have not collected & analyzed fs TAS data in this work.*

Response: We appreciate the reviewer's comment regarding the characterization of the molecular triplet exciton as a "dark state." We agree with the reviewer that the term "dark state" can be misleading as triplet excitons are not entirely inaccessible to light. Triplet states predominantly decay through non-radiative processes or much slower radiative processes like phosphorescence. In our study, we referred to the triplet state as the "dark state" to distinguish the non-fluorescent :PDI²⁻ isomer from the fluorescent PDI²⁻ quinoids, as shown in **Figure 1e** in the manuscript. To provide a more accurate description, we have added further explanations to this statement in the revised manuscript (Page 6, Line 18~19).

In response to the reviewer's suggestion, we have conducted femtosecond transient absorption spectroscopy (fs-TAS) experiments on the :PDI²⁻/PDI²⁻ samples before and after 120 h of self-assembly (**Figure R2**). The 120 h self-assembled :PDI²⁻/PDI²⁻ sample showed a rapid singlet fission (SF) process (occurring in 300 fs) and the emergence of a triplet excited state absorption (ESA) feature (around 770 nm) after SF upon $\lambda_{\text{ex}} = 540$ nm excitation (**Figure R2d-g**). This observation contrasts with the behavior of the :PDI²⁻/PDI²⁻ molecular precursor (**Figure R2a-c**). An identical triplet ESA at ~770 nm, resulting from the direct S_0-T_1 transition, was also observed under $\lambda_{\text{ex}} = 700$ nm excitation (**Figure R2h-j**). The fs-TAS findings are consistent with our photoluminescence (PL) emission experiments and further support our proposed mechanism. For a detailed data analysis, please refer to Page 12, Line 30~31 and Page 13, Line 1~10 in the revised manuscript.

Figure R2. Femtosecond transient absorption: 2D map (left), spectral traces for selected time point (middle) and TA kinetic traces extracted under different wavelength (right) **(a)-(c)** :PDI²⁻/PDI²⁻ (0 h), $\lambda_{\text{ex}}=540$ nm, **(d)-(g)** :PDI²⁻/PDI²⁻ (120 h), $\lambda_{\text{ex}}=540$ nm and **(h)-(j)** :PDI²⁻/PDI²⁻ (120 h), $\lambda_{\text{ex}}=700$ nm.

Q3: *The maximum quantum yield reported by the authors (1.5%) is extremely low and very far from practical use. The weak water-splitting performance also has consequences for the science described in this work, in the sense that a low quantum yield water-splitting could be generated through a regular singlet exciton generation and dissociation process without the need to invoke singlet fission or triplet fusion.*

Response: We appreciate the reviewer's comment regarding the quantum yield for our all-organic semiconductor catalyst. We understand the concerns about the quantum yield, and we are grateful for the opportunity to clarify this point. Our study presents the first all-organic, metal-free system for overall water splitting. To date, conventional organic systems with singlet excitation and electron-hole separation have been incapable of splitting water into H₂ and O₂ without the assistance of noble metal loading. This limitation mainly arises because the hydrogen evolution overpotential for organic materials is too high to operate under visible light irradiation.

Our unique self-assembled :PDI²⁻/PDI²⁻ nanobelt system overcomes this challenge through a singlet

fission process followed by triplet fusion. This approach effectively transforms short-lived singlet excitons into two long-lived triplet excitons while concentrating the excitation energy on the tautomeric PDI²⁻ quinoid isomer, thereby enabling H₂ evolution. **Table R1** summarizes the maximum AQY of recently published organic material/inorganic non-metallic materials-induced *overall* water splitting. Within this context, our :PDI²⁻/PDI²⁻ nanobelt achieves a maximum apparent quantum efficiency (AQE) of 1.98% at 550 nm and an average AQE of ~1.5% from 500-800 nm, which is competitive among these materials. Notably, our catalyst is the only all-organic system capable of achieving overall water-splitting without the need for expensive or hazardous metal components.

Moreover, the direct S_0-T_1 excitation and triplet fusion up-conversion (TFU) enabled near-infrared (NIR) light-driven water splitting with an AQY of ~1% at 850 nm, a region of solar spectra that has traditionally been considered energy-insufficient. Most importantly, the SF-TFU-resulted *S₁ final state is more sensitive to water than the normal S_0-S_1 excitation on the spin-hybrid :PDI²⁻/PDI²⁻ nanobelt (**Figure 4f**), necessitating the singlet fission (SF)-TPU exciton transition route in our system.

While we acknowledge that the current efficiency is far from ideal for practical applications, our catalyst serves as an important proof-of-concept for further research and optimization in this field. Future studies can focus on enhancing the efficiency and stability of similar organic photocatalysts, potentially leading to more practical and environmentally-friendly applications in the realm of water splitting.

Table R1. A summary of recent published organic/inorganic nonmetallic materials-induced overall water splitting with the H₂/O₂ production rate, light source and maximum AQYs.

Photocatalyst	Co-catalyst	Gas evolution rate		Light source	Maximum AQY (%)	Ref
		(μmol g ⁻¹ h ⁻¹)				
		H ₂	O ₂			
PDI/Zn _{0.8} Cd _{0.2} S	/	71.98	32.44	Xe lamp	/	1
SWCNT/C ₃ N ₄	Pt and Ru	49.8	22.8	Xe lamp	/	2
TH-CN	CoP and Pt	204	114	Xe lamp	/	3
aza-CMP/rGO/C ₂ N	Pt and Co(OH) ₂	1600	800	Xe lamp	4.3 (λ=600 nm)	4
g-C ₃ N ₄ /rGO/PDIP	Pt/Cr ₂ O ₃ and Co(OH) ₂	632	312	Xe lamp	4.94 (λ=420 nm)	5
P10	Ir and Pd	740	140	Xe lamp	/	6
PTI·HCl	Pt and Co	650	325	Xe lamp	2.6 (λ=365 nm)	7
TpBpy-NS	Pt	660	320	Xe lamp	2.8 (λ=450 nm)	8
F-CN	Pt	177.79	46.47	Xe lamp	0.5718 (λ=365 nm)	9
:PDI ²⁻ /PDI ²⁻	/	409.7	171.2	Xe lamp	1.96 (λ=550 nm)	This work

References

1. Liu Z. et al. Synthesis of PDI/Zn_{0.8}Cd_{0.2}S composites for efficient visible light-driven photocatalytic overall water splitting. J. Taiwan Inst. Chem. E. **143**, 104693 (2023).
2. Wang S. et al. Efficient photocatalytic overall water splitting on metal-free 1D SWCNT/2D ultrathin C₃N₄ heterojunctions via novel non-resonant plasmonic effect. Appl. Catal. B: Environ. **278**, 119312 (2020).
3. Pan Z. et al. Molecular triazine–heptazine junctions promoting exciton dissociation for overall water splitting with visible light. J. Phys. Chem. C. **125**, 9818-9826 (2021).
4. Wang L. et al. Van der waals heterostructures comprised of ultrathin polymer nanosheets for efficient z-scheme overall water splitting. Angew. Chem. Int. Ed. **57**, 3454-3458 (2018).
5. Chen X. et al. Efficient photocatalytic overall water splitting induced by the giant internal electric field of a g-C₃N₄/rGO/PDIP Z-scheme heterojunction. Adv. Mater. **33**, 2007479 (2021).
6. Bai Y. et al. Photocatalytic overall water splitting under visible light enabled by a particulate conjugated polymer loaded with palladium and iridium. Angew. Chem. **134**, e202201299 (2022).
7. Lin L. et al. Photocatalytic overall water splitting by conjugated semiconductors with crystalline poly(triazine imide) frameworks. Chem. Sci. **8**, 5506-5511 (2017).
8. Yang Y. et al. Engineering β-ketoamine covalent organic frameworks for photocatalytic overall water splitting. Nat. Commun. **14**, 593-603 (2023).
9. Wu J. et al. Breaking through water-splitting bottlenecks over carbon nitride with fluorination. Nat. Commun. **13**, 6999-7007 (2022).

Q4: *H₂ and O₂ are claimed to be generated in a perfect 2:1 stoichiometric ratio, but Figure 3a shows otherwise. In fact, the ratio varies from 2.3 to 3. Why?*

Response: We appreciate the reviewer's comment regarding the H₂/O₂ ratio in **Figure 3a**. It is true that the displayed ratio does not perfectly match the expected 2:1 stoichiometry. In practical overall water-splitting experiments, achieving an ideal H₂/O₂ ratio of 2:1 can be challenging due to factors such as the dissolution of O₂ and the inevitable production of H₂O₂ (1, 2). In our previous studies, we have reported several inorganic catalysts with H₂/O₂ ratios of 3:1 or 4:1 (3-5). In these cases, excessive holes were primarily consumed by H₂O₂ production.

In the present work, we also detected H₂O₂ production on our :PDI²⁻/PDI²⁻ catalyst (**Figure R3**). The H₂O₂ production rate is approximately 60 μmol·g⁻¹·h⁻¹, which accounts for the discrepancy in the H₂/O₂ ratio deviating from the ideal 2:1 ratio (409/171 → 409/201 (171+30)).

We have included additional descriptions in the revised manuscript to address this point and provide further clarification. Please refer to Page 10, Line 10~12.

Figure R3. Time-profile of H₂O₂ production during the overall water-splitting experiment on :PDI²⁻/PDI²⁻ (120 h) catalyst.

References

1. Wang M. et al. Atomically dispersed janus nickel sites on red phosphorus for photocatalytic overall water splitting. *Angew. Chem. Int. Ed.* **61**, e202204711 (2022).
2. Fu Y. Photocatalytic H₂O₂ and H₂ generation from living chlorella vulgaris and carbon micro particle comodified g-C₃N₄. *Adv. Energy Mater.* **8**, 1802525 (2018).
3. Jiang E. et al. Nanochannel-induced efficient water splitting at the superhydrophobic. *Interface. ACS Nano.* **17**, 10774-10782 (2023).
4. Wu J. et al. Breaking through water-splitting bottlenecks over carbon nitride with fluorination. *Nat. Commun.* **13**, 6999 (2022).
5. Jiang E. et al. Vertical growth of O-vacancy rich LDH atomic layers as OER-sensitive reactive sites to boost overall water splitting on perovskite oxides. *ACS Sustain. Chem. Eng.* **10**, 16335-16343 (2022).

Q5: *LACK OF QUANTIFICATION: The yields of the intersystem crossing process, singlet fission process and triplet fusion process are not provided.*

Response: We appreciate the reviewer's comment. As suggested, we have determined the yields for singlet fission (SF) and triplet fusion using fs-TAS and PL quantum efficiency measurements. Due to the large energy gap between singlet and triplet states ($\Delta E > 1.44$ eV), intersystem crossing is unlikely to occur in our system.

For the 120-hour self-assembled PDI/PDI nanobelt sample, we calculated the SF efficiency to be 42%.

Additionally, the triplet-triplet annihilation (TTA) upconversion emission quantum efficiency ($\lambda_{\text{ex}} = 450 \text{ nm}$, $\lambda_{\text{em}} = 719 \text{ nm}$) was determined to be 0.688%. Detailed data analysis and further information can be found in the revised manuscript on Page 12, Line 11~12 and Page 13, Line 10~15.

OTHER SCIENTIFIC QUESTIONS

Q6: *In Figure 1a and 1d, the absorption and PL spectra of :PDI²⁻/PDI²⁻ look too similar, suggesting a negligible Stokes shift, which would be unusual in organic semiconductors that tend to exhibit a large reorganization energy. This aspect requires more comment and elaboration. It is highly recommended that the authors collect Raman spectra from the sample, since this data provided an independent method to obtain the reorganization energy.*

Response: We appreciate the reviewer's comment regarding the absorption and PL spectra of :PDI²⁻/PDI²⁻ and the associated Stokes shift. Indeed, compared to PDI, :PDI²⁻/PDI²⁻ exhibits a much smaller Stokes shift (**Figure R4**), which can be distinguished as ~15 nm, suggesting a smaller reorganization energy than that of PDI. Although the reorganization energy cannot be directly measured, differences in molecular vibrational modes upon excitation can be detected and distinguished using in-situ Raman and IR spectra.

As requested, we first performed in-situ confocal Raman spectroscopy on the sample. However, we encountered significant interference from the PL emission of the sample upon excitation (**Figure R5**). Consequently, we conducted in-situ IR spectroscopy on both :PDI²⁻/PDI²⁻ and PDI upon excitation (Xe lamp). As shown in **Figure R6**, compared to PDI, :PDI²⁻/PDI²⁻ demonstrates smaller changes and shifts in infrared vibrational intensity under the same excitation conditions, indicating a smaller reorganization energy. This observation is consistent with the smaller Stokes shift of :PDI²⁻/PDI²⁻.

We have included a description of these findings in the revised manuscript. Please refer to Page 6, Line 14~17.

Figure R4. UV-vis absorption and PL emission spectra of (a) PDI and (b) :PDI²⁻/PDI²⁻. The Stokes shift is marked.

Figure R5. In situ laser confocal Raman spectra with different percentage of incident light intensity (λ_{ex}=532 nm) on (a) PDI and (b) :PDI²⁻/PDI²⁻.

Figure R6. In-situ FT-IR spectra of (a) PDI and (b) $\text{:PDI}^{2-}/\text{PDI}^{2-}$ under constant white-light (300W Xe-lamp) irradiation.

Q7: The authors claim that $\text{:PDI}^{2-}/\text{PDI}^{2-}$ is soluble in water. Photographs and spectra of the aqueous solutions should be provided, at least in Supporting Information.

Response: We appreciate the reviewer's comment. In our manuscript, the photographs and spectra of PDI and $\text{:PDI}^{2-}/\text{PDI}^{2-}$ aqueous solutions were presented in **Figure 1a** and the corresponding inset. To provide further clarification, we have included a comparison of photographs of PDI and $\text{:PDI}^{2-}/\text{PDI}^{2-}$ aqueous solutions in the Supporting Information (**Supplementary Figure 1**).

Figure R7. Photographs of 0.04 g L^{-1} PDI and $\text{:PDI}^{2-}/\text{PDI}^{2-}$ aqueous solution.

Q8: Upconversion processes are known to be nonlinear. Therefore it is really important that the authors measure & report the emission quantum yields & lifetimes of the upconversion luminescence as a function of excitation intensity over at least 3 orders of magnitude of intensity. In other words, I am asking for Figures 4c -to 4f to improved and augmented.

Response: We appreciate the reviewer's comment on the nonlinear nature of upconversion processes. It is true that traditional two-photon or multi-photon upconversion processes typically exhibit nonlinear behavior. However, in our system, the triplet fusion upconversion (TFU) occurs immediately following singlet fission (SF), which differs from the conventional upconversion mechanisms. As a result, the degree of nonlinearity in our system may be distinct from other systems.

In response to the reviewer's request, we performed additional steady-state and transient PL emission spectra measurements with varying light intensities ($10 \mu\text{J cm}^{-2}$ - $30 \mu\text{J cm}^{-2}$). As shown in **Figure R8a**, the TTA upconversion emission intensity at 713 nm increases with the increasing incident light intensity, showing a linear relationship with the incident light intensity (**Figure R8b**). This observed linearity does not conform to the typical two-photon upconversion mode (**Figure R8b**), which is consistent with the SF-TFU transition route. The SF and TFU processes are inversely related and interconnected, leading to a less pronounced nonlinear behavior. In this SF-TFU route, the SF process generates two triplets from one singlet state, and the subsequent TFU process combines two triplets to generate one higher-energy singlet state, emitting a photon (if not to react with H_2O) as a result. This interconnected process results in a more linear relationship between excitation intensity and emission, contrasting with conventional upconversion mechanisms. Similar phenomenon has been observed on other organic photovoltaic cells that employs a pentacene-based SF process with a sublinear dependence on excitation intensity, which is distinct from the typical nonlinear response in conventional multi-photon upconversion systems. (*Science* 340, 334-337 (2013)).

Notably, the lifetime of the upconversion emission remains unchanged with different incident light intensities, suggesting that no saturation or intensity-induced quenching (such as exciton-exciton annihilation or energy pooling) occurs during this process. We have included all these additional data in the revised manuscript; please refer to Page 12, Line 12~24.

Figure R8. (a) Intensity dependent PL spectra ($\lambda_{\text{ex}}=450$ nm) on :PDI²⁻/PDI²⁻ (120 h) sample ($10 \mu\text{J cm}^{-2}$ - $30 \mu\text{J cm}^{-2}$); (b) The linear relationship between integrated PL intensity and incident light intensity (and the square I^2); (c) Intensity dependent transient PL decay profile of $\lambda_{\text{em}}=713$ nm emission ($\lambda_{\text{ex}}=450$ nm) on :PDI²⁻/PDI²⁻ (120 h).

To Reviewer 3:

Comments:

Lin et al. reported a direct solar-to-hydrogen conversion from pure water using an all-organic heterogeneous catalyst. Although the results sound interesting, the current study is incomplete and has many flaws and errors. Therefore, the current version cannot be published in Nature Communications, and authors should be encouraged to do more work to prove whether the work is suitable for publication in the future. The details are as follows:

Q1: *The authors obtained PDI²⁻ by introducing a strong alkaline reducing agent, and PDI²⁻ is not stable in air, which greatly weakens the highlight of the article and reduces the real application of the material in water-splitting reaction. The authors should consider how to make PDI²⁻ relatively stable in the air or in-situ formation rather than by chemical reduction.*

Response: We appreciate the reviewer's comment and acknowledge the reviewer's concern about the stability of PDI²⁻ in air and in water-splitting reactions. First, our study is a proof-of-concept work, which demonstrates

an innovative mechanism for achieving overall water splitting using organic materials with an entirely non-metallic, near-infrared (NIR) response. This pioneering work lays the foundation for future catalyst design and system optimization to improve material stability.

Second, we have recently made progress based on this concept, addressing some of the concerns raised by the reviewer. For instance, we have developed a $\text{Zn}_{0.5}\text{Cd}_{0.5}\text{S}/\text{PDI}$ composite system, in which $:\text{PDI}^{2-}$ radicals are in-situ formed during the reaction process. The $:\text{PDI}^{2-}$ radicals can be produced by $\text{Zn}_{0.5}\text{Cd}_{0.5}\text{S}$ quantum dots photoreduction, driving the reaction without the need for pre-assembly, and achieving S_0-T_1 NIR absorption, as shown in the provided **Figure R9a, b**. This approach addresses the issue of $:\text{PDI}^{2-}$ stability by generating the radical species in-situ during the reaction.

Furthermore, we have prepared PDI-immobilized films with robust chemical binding on the film surface (**Figure R9c, d**). After chemical reduction, the reaction can be driven without self-assembly, and the catalyst can be preserved in its PDI form upon reaction completion. This approach allows for the stable storage of the catalyst and addresses the stability concerns raised by the reviewer.

In conclusion, we believe that our proof-of-concept study demonstrates a promising mechanism for achieving complete water splitting using organic materials, with future work focused on improving material stability and system optimization. We appreciate the reviewer's insightful comments.

Figure R9. (a) Photographs of $0.4 \text{ g L}^{-1} \text{Zn}_{0.5}\text{Cd}_{0.5}\text{S}/\text{PDI}$ aqueous solution during irradiation by 300 W Xe lamp in 30 min, and then expose to air for 10 min, (b) In-situ UV-vis absorption spectra of $\text{Zn}_{0.5}\text{Cd}_{0.5}\text{S}/\text{PDI}$, (c) Photographs of PDI membrane before and after hydrazine hydrate reduction, and then expose to air for 10 min, and (d) UV-vis absorption spectra of different membrane samples.

Q2: In Figure 1e, PDI could be reduced by N_2H_4 within 30 min; the fluorescent peak at 650 nm could be attributed to the PDI^{2-} . This is undoubtedly true because a similar phenomenon has been found in the reported

article (*Angew. Chem. Int. Ed.* 2022, 61, e202110491). However, once the PDI^{2-} was transferred to the DMF, the fluorescent peak at 650 nm showed a blueshift to 540 nm, just like the PDI (Figure 1f). The authors still attribute this peak to PDI^{2-} . This is a completely wrong conclusion; PDI has been oxidized at this time, and the author needs to reconsider this conclusion.

Response: We appreciate the reviewer's comment and apologize for any misunderstanding caused by the presentation of our data. We have revised our explanation to provide a clearer understanding of the processes depicted in **Figure 1e**. As described in the manuscript, Figure 1e demonstrates two different processes, which we have separated into two new figures below (**Figure R10**). In DMF solutions, upon the addition of reducing agent $\text{N}_2\text{H}_4\cdot\text{H}_2\text{O}$, the fluorescence peak of PDI rapidly decreases to zero within 10 seconds, resulting in the formation of non-fluorescent $:\text{PDI}^{2-}$ triplet radical anions (**Figure R10a**). However, within the next 30 minutes, fluorescence peaks at 646 nm and 690 nm gradually emerge, indicating the transition from non-fluorescent $:\text{PDI}^{2-}$ to fluorescent PDI^{2-} via tautomerism transformation (**Figure R10b**). That is, the tautomerism transformation from reduced diradical $:\text{PDI}^{2-}$ to quinoid isomer PDI^{2-} reaches an equilibrium (75% to 25%) within a 30-minute timeframe, and the quinoid isomer PDI^{2-} exhibits fluorescence.

It is important to note that the observed fluorescence peaks of $:\text{PDI}^{2-}/\text{PDI}^{2-}$ in DMF in the presence of $\text{N}_2\text{H}_4\cdot\text{H}_2\text{O}$ are significantly redshifted compared to the solid $:\text{PDI}^{2-}/\text{PDI}^{2-}$ sample (**Figure 1d**). To confirm this redshift originates from the DMF solvent or the presence of $\text{N}_2\text{H}_4\cdot\text{H}_2\text{O}$, we conducted control experiments as depicted in **Figure 1f**. We extracted $:\text{PDI}^{2-}/\text{PDI}^{2-}$ solid from the solution (dried under N_2) and, redissolved it in DMF, and observed a substantial blueshift in PL spectra. The final fluorescence profile is consistent with the solid $:\text{PDI}^{2-}/\text{PDI}^{2-}$ sample. Importantly, this process was conducted in a strictly anaerobic environment, making oxidation of $:\text{PDI}^{2-}/\text{PDI}^{2-}$ impossible. Furthermore, the final fluorescence profile of $:\text{PDI}^{2-}/\text{PDI}^{2-}$ in DMF is distinct from the fluorescence of PDI in the same solvent (**Figure 1f**). We have updated **Figure 1e** and **Figure 1f** in the revised manuscript to improve clarity. Please see Page 6, Line 19~23 for the revised figures and explanations. We hope these revisions address the reviewer's concerns and provide a better understanding of our findings.

Figure R10. (a) PL spectra ($\lambda_{\text{ex}}=450$ nm) tracking the emission of PDI after the $2e^-$ reduction by hydrazine hydrate. With the addition of $0.25 \mu\text{L}$ hydrazine hydrate (85%), (b) In-situ PL spectra ($\lambda_{\text{ex}}=450$ nm) of $:\text{PDI}^{2-}$. A tautomeric equilibrium from the non-fluorescent diradicals to fluorescent quinoids was then achieved in 30 min.

Q3: During self-assembly, the authors believe that PDI^{2-} is assembled and the rod-like material is the PDI^{2-} . But according to our experiments, the same assembly can occur with PDI. Electron microscopy cannot distinguish between PDI^{2-} and PDI. In addition, PDI is an acid, and its assembly structure is destroyed while being reduced by an alkali-reducing agent, so a primitive disorder occurs. When placed in an aqueous solution of alcohol, it is reassembled, which is not a novelty and is not unique to PDI^{2-} .

Response: We appreciate the reviewer's comment and understand the concerns raised about the self-assembly of PDI^{2-} and its differentiation from PDI. We acknowledge that the self-assembly of PDI is a well-known feature and that the diradical $:\text{PDI}^{2-}/\text{PDI}^{2-}$ precursor, with its similar molecular framework, should also exhibit self-assembly behavior. However, we found two unique and unexpected aspects of the self-assembly of diradical $:\text{PDI}^{2-}/\text{PDI}^{2-}$ that differentiate it from the simple dispersion and reassembly of PDI:

i). Distinct crystalline structure: The self-assembled diradical $:\text{PDI}^{2-}/\text{PDI}^{2-}$ exhibits a completely different crystalline structure compared to PDI, despite having an identical molecular structure. The arrangement configuration changes from the face-to-face H-type to the head-to-tail J-type, suggesting a different driving force behind the self-assembly process.

ii). Triplet spin-coupling: Following the assembly of $:\text{PDI}^{2-}/\text{PDI}^{2-}$, a triplet spin-coupling occurs, which is unique to the diradical $:\text{PDI}^{2-}/\text{PDI}^{2-}$ self-assembly. This spin-coupling results in TTA-upconversion fluorescence emission that changes with the self-assembly time, as well as a direct S_0-T_1 transition under NIR absorption. These phenomena have not been observed in self-assembled PDI structures.

In summary, while the self-assembly of PDI^{2-} and PDI may share some similarities, there are key differences in the resulting crystalline structures and the presence of triplet spin-coupling in the

diradical ${}^{\cdot}\text{PDI}^2/\text{PDI}^{\cdot 2}$ self-assembly. These unique features distinguish the self-assembly behavior of diradical ${}^{\cdot}\text{PDI}^2/\text{PDI}^{\cdot 2}$ from that of PDI, adding novelty and significance to our findings.

Q4: *The isotope tracing experiments the authors carried out are also not credible. Firstly, the authors describe that the details of the isotope labeling experiment are extremely unclear, and there is no way to obtain reliable data for such a crude experiment. Secondly, Figure 3d (left) mentions the MS could be obtained from GC-MS. The carrier gas of GC-MS is He, and the mass-to-charge ratio of He is 4 and under the action of an ion source (ionization process), deuterium can react with residual oxygen in the mass spectrum (there is no absolute vacuum on Earth) to produce D_2O . So, it doesn't make sense to provide such a mass spectrum. Thirdly, Figure 3d (right) the authors mention using the injection method to inject samples. How did the authors prevent oxygen from the air from entering GC-MS? Based on our experiments, neither airtight needles (oxygen in the tip of the needle) nor headspace injection can avoid oxygen interference in the experiment. The author must give a reasonable explanation for these unreasonable results.*

Response: We appreciate the concerns raised by the reviewer regarding the credibility of our isotope tracing experiments. We would like to address each point raised and provide clarification on the experimental details.

First, we apologize that the isotope labeling experiment conditions were not described in detail in the manuscript. We provided more information and clarify each experimental step upon request to ensure the reliability of our data. Please see Page 18, Line 10~17.

Second, for the GC-MS carrier Gas and He interference: It is true that the carrier gas of GC-MS is He, and the mass-to-charge ratio (M/Z) of He is 4. However, He (24.6 eV) has a higher ionization energy than D_2 (15.4 eV). In our experiments, the electron beam energy was set at 20 eV, which does not ionize He, effectively avoiding its interference. Furthermore, due to the dilution effect, He as a carrier gas does not impact the D_2 measurement. To further verify this, we conducted a blank N_2 experiment and found that using He as a carrier gas at 20 eV does not produce a signal at M/Z= 4 for the N_2 blank sample (**Figure R11a**).

Third, for the sample injection and oxygen interference: Our gas samples were stored in gas bags, so the presence of air was inevitable. However, to accurately compare the ${}^{18}\text{O}_2$ and ${}^{16}\text{O}_2$ ratios, we deducted the ${}^{16}\text{O}_2$ -signal based on the N_2 signal, assuming the N_2/O_2 ratio in air is 78:21. This allowed us to obtain accurate ${}^{16}/{}^{18}\text{O}_2$ ratios, and the processed signal did not show N_2 (**Figure 11c**). The raw data, including N_2 , is presented below in **Figure R11b** and added to the revised manuscript.

Figure R11. GC-MS profiles of (a) N₂ blank sample (no He background signal detected), and isotopically labeled gas products from water-splitting over :PDI²⁻/PDI²⁻ (120 h) in deuterium-labeled ¹⁸O-labeled H₂¹⁸O: (b) raw data and (c) calibrated by deducting air. All experiments were conducted with electron beam energy of 20 eV.

REVIEWER COMMENTS

Reviewer #1 (Remarks to the Author):

The revised version is ready for publication.

Reviewer #3 (Remarks to the Author):

In this round, the authors tried to respond to the concerns of reviewers, but I am still not satisfied with the answers, and the manuscript did not meet the requirements for publication. Details are as follows:

1. The author introduce the ZnCdS₂ as the photoelectron donor to produce the :PDI₂⁻, however this photocatalyst could act as the donor in the presence of another electron donor. The auhtor did not described the process in detail.
2. The authors still attribute the peak at 540 and 570 to :PDI₂⁻/PDI₂⁻ which is also most the same as the PDI. There is no way to understand this, and the explanation here is not clear, and there must be other reasons behind it that have not been discovered.
3. The authors claim that the structure of PDI₂⁻ is different from the PDI⁻. Please provide crystal parameters and electron diffraction data as important support.
4. The isotope experimtet is the most serious problem in the article. a) The presence of multiple substances in the same mass spectrum indicates that the substances are not separated in the chromatography, which will greatly interfere with the detection of the product, so the traceability results under these conditions are completely meaningless. b) We admit that He can not be ionized under the premise of adjusting the ionization energy, but the ionization degree of N₂ and O₂ is also affected at this time, so the ratio of the two is definitely not 78:21 as claimed by the author. The reviewer therefore seriously doubts the authenticity of this data, and if the detailed raw data is not provided, the manuscript will be rejected. c) Both H₂ and D₂ could react with O₂ (cannot be ruled out) during ionization. The author claimed they use gas bags for sampling to exclude the air, it is impossible, the authors are request to provide the raw data of the TIC and MS (In the form of a clear photo).

Reviewer #5 (Remarks to the Author):

This work provides a new systematical analysis for the performance of the :PDI₂⁻/PDI₂⁻ catalyst for visible-light driven overall water splitting process from the photochemsitry insights, which is very nice. However, there are still some details to be concerned.

1. For the PDI₂⁻ quinoid isomer, the singlet photoluminescence emissions at 540 nm and 585 nm, shown in Fig. 2d, while the corresponding fluorecence peaks are at 646 nm and 690 nm (Fig.2e) via tautomerism transformation. What makes the differences?

2. Why could the stock shift of UV-vis absorption and PL emission spectra suggest the lower reorganization energy, please give some references.
3. How to keep high stability of the :PDI2-/PDI2- nonobelt in the acid solution for driving HER?
4. Direct S0→T1 transition is ascribed to spin-orbit coupling, please give more details or references about the mechanism.
5. How to evaluate the charge separation degree within the :PDI2-/PDI2- organic semiconductor?

Reply to reviewers' comments

To Reviewer 3:

Comments:

In this round, the authors tried to respond to the concerns of reviewers, but I am still not satisfied with the answers, and the manuscript did not meet the requirements for publication. Details are as follows:

Q1: *The author introduce the ZnCdS₂ as the photoelectron donor to produce the :PDI²⁻, however this photocatalyst could act as the donor in the presence of another electron donor. The author did not described the process in detail.*

Response: We appreciate the reviewer's concerns regarding details for the activation process of our newly developed Zn_{0.5}Cd_{0.5}S/PDI composite system. In this case, Zn_{0.5}Cd_{0.5}S indeed acts as a primary photoelectron donor to reduce PDI into diradical :PDI²⁻ by a 2e⁻-transfer manner, but it is still essentially a photocatalyst (see **Scheme R1**). Under light irradiation, Zn_{0.5}Cd_{0.5}S absorbs the light and generates electron-hole pairs. The photogenerated electrons are then transferred to PDI to produce :PDI²⁻. Meanwhile, the photogenerated holes on the valence band of Zn_{0.5}Cd_{0.5}S require a timely external electron donor for consumption. For instance, in a pure water system, water molecules serve as the final electron donor. The Zn_{0.5}Cd_{0.5}S/PDI composite only catalyzes the oxidation of the water molecules, leaving extra electrons/protons into PDI and generating diradical :PDI-H₂ (**Figure R1a-1c**). A similar process can also be observed when triethanolamine (TEOA) is used as the organic electron donor (**Figure R1d-1f**). In other words, we take advantage of the charge transfer between the Zn_{0.5}Cd_{0.5}S/PDI heterojunction and the oxidation of the final molecular electron donor to in-situ generate triplet diradical :PDI²⁻ during the photolysis process. Here, Zn_{0.5}Cd_{0.5}S exhibits unique 2e⁻-transfer features that allow it to reduce the bonded PDI moiety into :PDI²⁻ rather than producing a single free radical PDI[•] product of 1e⁻-transfer. We hope the additional experimental details and explanation bring clarity to the activation process of our newly developed Zn_{0.5}Cd_{0.5}S/PDI composite system.

Scheme R1. Schematic diagram of the photoactivation process in $\text{Zn}_{0.5}\text{Cd}_{0.5}\text{S}/\text{PDI}$ composite.

Figure R1. (a) Photographs of 0.4 g L^{-1} $\text{Zn}_{0.5}\text{Cd}_{0.5}\text{S}/\text{PDI}$ aqueous solution during irradiation by 300 W Xe lamp in 30 min. (b) In-situ UV-vis absorption spectra of $\text{Zn}_{0.5}\text{Cd}_{0.5}\text{S}/\text{PDI}$ in H_2O under continuous irradiation. (c) In-situ ESR spectra of $\text{Zn}_{0.5}\text{Cd}_{0.5}\text{S}/\text{PDI}$ in H_2O under continuous irradiation. (d) Photographs of 5 mg $\text{Zn}_{0.5}\text{Cd}_{0.5}\text{S}/\text{PDI}$ in 20 mL TEOA solution during irradiation by 300 W Xe lamp in 30 min. (e) In-situ UV-vis absorption spectra of $\text{Zn}_{0.5}\text{Cd}_{0.5}\text{S}/\text{PDI}$ in 10 wt% TEOA aqueous solution under continuous irradiation. (f) In-situ ESR spectra of $\text{Zn}_{0.5}\text{Cd}_{0.5}\text{S}/\text{PDI}$ in TEOA under continuous irradiation.

Q2: The authors still attribute the peak at 540 and 570 to $\text{:PDI}^{2-}/\text{PDI}^{2-}$ which is also most the same as the PDI.

There is no way to understand this, and the explanation here is not clear, and there must be other reasons behind it that have not been discovered.

Response: We apologize for any confusion stemming from our initial explanation regarding the photoluminescence (PL) emission spectra changes of the formed $\text{:PDI}^{2-}/\text{PDI}^{2-}$ tautomer upon the addition of hydrazine hydrate. To clarify, let's first consider the individual luminescence spectra of three species each, namely, PDI, :PDI^{2-} diradical, and its tautomer PDI^{2-} , in DMF solutions (refer to **Figure R2a-c**). The precursor PDI in DMF has three PL peaks (at 542, 577 and 630 nm) (**Figure R2a**). The isolated reduced PDI^{2-} quinoid isomer has two PL peaks at 540 nm and 585 nm (**Figure R2c**). However, the triplet diradical :PDI^{2-} of the tautomer $\text{:PDI}^{2-}/\text{PDI}^{2-}$ mixture is non-radiative throughout (**Figure R2a**). A notable difference exists in the PL spectra between PDI and the reduced tautomer $\text{:PDI}^{2-}/\text{PDI}^{2-}$ mixture. Moreover, the PL characteristics of PDI^{2-} in different solutions can be significantly influenced by the co-existence of other compounds, such as hydrazine hydrate or its oxidized products. Thus, it's understandable that the addition of hydrazine hydrate causes the peaks of PDI^{2-} to redshift significantly from 540 nm and 585 nm to 641 nm and 688 nm, respectively (see **Figure R2b**). In other words, the presence of extra hydrazine hydrate or its oxidized products significantly affects the fluorescence features of the $\text{:PDI}^{2-}/\text{PDI}^{2-}$ tautomer mixture in solution. In our initial response, we compared the PL emission spectra of $\text{:PDI}^{2-}/\text{PDI}^{2-}$ after reduction with residual hydrazine hydrate and isolated $\text{:PDI}^{2-}/\text{PDI}^{2-}$ without hydrazine hydrate. We attributed the fluorescence changes to the presence of hydrazine hydrate in solution.

To further validate this, we performed additional control experiments. We added varying amounts (0~0.35 μL) of hydrazine hydrate to the DMF solution of the isolated $\text{:PDI}^{2-}/\text{PDI}^{2-}$ tautomer mixture, which was separated from the original reduction system of PDI with hydrazine hydrate by heating and evaporating the solvent in an oxygen-free environment. As shown in **Figure R2d**, we observed two distinct peaks at 540 nm and 585 nm-attributed to the fluorescence emission of PDI^{2-} -before adding hydrazine hydrate on the isolated $\text{:PDI}^{2-}/\text{PDI}^{2-}$ sample in DMF. A gradual redshift in the PL emission of $\text{:PDI}^{2-}/\text{PDI}^{2-}$ was observed with the addition of hydrazine hydrate. This strongly supports that the presence of hydrazine hydrate in solution indeed caused a significant redshift in the fluorescence of $\text{:PDI}^{2-}/\text{PDI}^{2-}$. This outcome may be due to the coordination interaction between the alkaline hydrazine hydrate molecules and the carboxyl groups of the $\text{:PDI}^{2-}/\text{PDI}^{2-}$ tautomer molecules, a phenomenon that is a common event (1, 2). Further, when we introduced other alkaline small molecules such as ammonia, DEA, and TEOA into the DMF solution of $\text{:PDI}^{2-}/\text{PDI}^{2-}$, we observed similar significant changes and a redshift in the PL emission of $\text{:PDI}^{2-}/\text{PDI}^{2-}$ (See **Figure R2e**). This supports the idea that coordination interaction causes changes in the fluorescence of $\text{:PDI}^{2-}/\text{PDI}^{2-}$ molecules. We hope these additional experimental details and references provide a clearer understanding of

this process. Additional experimental results have been added to the revised manuscript, please see **Page 6**, **Lines 26~32**, and **Supplementary Fig. 10**.

Figure R2. (a) In-situ PL spectra ($\lambda_{\text{ex}}=450$ nm) tracking the emission of PDI after the $2e^-$ reduction by hydrazine hydrate in DMF solution in 30 min. With the addition of $0.25 \mu\text{L}$ hydrazine hydrate (85%), PDI was first converted to non-fluorescent $:\text{PDI}^{2-}$ diradicals. A tautomeric equilibrium from the non-fluorescent diradicals to fluorescent quinoids was then achieved in 30 min. (b) Comparing PL spectra ($\lambda_{\text{ex}}=450$ nm) between pristine PDI and $:\text{PDI}^{2-}/\text{PDI}^{2-}$ samples from in-situ $2e^-$ reduction of PDI for 30 min before and after removing residue hydrazine hydrate in DMF solutions. (c) The PL spectrum ($\lambda_{\text{ex}}=450$ nm) of $:\text{PDI}^{2-}/\text{PDI}^{2-}$ solid precursor sample. (d) PL spectra ($\lambda_{\text{ex}}=450$ nm) of isolated $:\text{PDI}^{2-}/\text{PDI}^{2-}$ DMF solution in addition of different amount of hydrazine hydrate. (e) PL spectrum ($\lambda_{\text{ex}}=450$ nm) spectra ($\lambda_{\text{ex}}=450$ nm) of isolated $:\text{PDI}^{2-}/\text{PDI}^{2-}$ DMF solution in addition of $0.35 \mu\text{L}$ ammonia, DEA, and TEOA.

References:

1. Yersin, H. et al. The triplet state of organo-transition metal compounds. Triplet harvesting and singlet harvesting for efficient OLEDs. *Coord. Chem. Rev.* 225, 2622-2652 (2011).

2. Zhang, G. et al. A dual-emissive-materials design concept enables tumor hypoxia imaging. *Nat. Mat.* 8, 747-751 (2009).

Q3: The authors claim that the structure of PDI^{2-} is different from the PDI. Please provide crystal parameters and electron diffraction data as important support.

Response: We appreciate the reviewer's suggestion. In this work, we prepared two kind organic crystals with different structures, namely, PDI and $\text{:PDI}^{2-}/\text{PDI}^{2-}$. As requested, the comparison of SAED images (**Figure R3**) and crystal parameters (**Table R1**) between PDI and $\text{:PDI}^{2-}/\text{PDI}^{2-}$ were provided in the revised manuscript. Please refer to **Supplementary Fig. 14** and **Supplementary Table 1**.

Figure R3. (a) Single-crystal SAED pattern of $\text{:PDI}^{2-}/\text{PDI}^{2-}$ nanobelts. (b) A simulated arrangement model of $\text{:PDI}^{2-}/\text{PDI}^{2-}$ crystal according to the SAED and XRD patterns. (c) Single-crystal SAED pattern of PDI. (d) A simulated arrangement model of PDI crystal according to the SAED and XRD patterns.

Table R1. Crystal parameters of PDI and $\text{:PDI}^{2-}/\text{PDI}^{2-}$.

Sample	length of a-axis	length of b-axis	length of c-axis
	(Å)	(Å)	(Å)
PDI	14.00	5.25	8.69
$\text{:PDI}^{2-}/\text{PDI}^{2-}$	16.80	8.29	6.40

Q4: *The isotope experiment is the most serious problem in the article.*

a) The presence of multiple substances in the same mass spectrum indicates that the substances are not separated in the chromatography, which will greatly interfere with the detection of the product, so the traceability results under these conditions are completely meaningless.

b) We admit that He can not be ionized under the premise of adjusting the ionization energy, but the ionization degree of N₂ and O₂ is also affected at this time, so the ratio of the two is definitely not 78:21 as claimed by the author. The reviewer therefore seriously doubts the authenticity of this data, and if the detailed raw data is not provided, the manuscript will be rejected.

c) Both H₂ and D₂ could react with O₂ (cannot be ruled out) during ionization. The author claimed they use gas bags for sampling to exclude the air; it is impossible, the authors are request to provide the raw data of the TIC and MS (In the form of a clear photo).

Response: We greatly appreciate the reviewer's meticulous feedback and constructive concerns about the GC-MS data. We would like to address each point raised:

a) For the concerns regarding the presence of multiple substances in the same mass spectrum. The GC-MS system used in our experiment separates analyte gases before they enter the mass spectrometer, and our data effectively demonstrates distinct retention times for D₂ (1.492 min) and ¹⁸O₂ (2.629 min). The full retention time profile of the total ion current (TIC) gas-chromatogram provides evidence of this separation and will be provided for reference (**Figure R4 b, d**). The presence of multi-substance signals in mass spectra was due to the introduction of air/N₂ during the sampling process and the challenges in separating N₂/O₂ by gas chromatography. We emphasize that the detection of ¹⁸O-labeled oxygen indicated the decomposition of ¹⁸O-labeled water (H₂¹⁸O) (**Figure R4c**), affirming that water decomposition indeed occurred.

b) For concerns about the N₂/O₂ ratio and the authenticity of our data. We have provided all raw data reports, including photographs of the TIC gas-chromatogram and mass spectra and corresponding CSV files (**Figure R4, R5, Supplementary files 1-3**), which are verifiable and ensure the authenticity of our results. We acknowledge that our earlier deducing by the N₂/O₂ (78:21) ratio was not scientifically quantitative as pointed out by the reviewer. Such an action was only a preliminary measure to highlight the proportion of ¹⁸O₂ in the product. According to the reviewer's suggestion, we have refrained from deducting air from the raw data, and reported the uncalibrated raw data figure (**Figure 3d**). In this work, GC-MS measurement is primarily for qualitative characterization of the decomposition products of isotopically-labeled water (D₂ and ¹⁸O₂, respectively). We believe that the observation of D⁺, HD, D₂ (m/z=2, 3 and 4) (**Figure R4a** and **Supplementary File 1**) and ¹⁶O¹⁸O, ¹⁸O₂ (m/z=34 and 36) (**Figure R4c** and **Supplementary File 2**) signals is sufficient to confirm that water decomposition has indeed occurred, supporting our conclusion.

c) For concerns about the possible back-reaction of H₂ with ¹⁸O₂ during ionization in the mass spectrometer. While we acknowledge the potential for these gases to react during ionization, this does not negate their initial presence. Considering our aim is to qualitatively detect the presence of these gases, not to measure their precise quantities, the potential for such reactions does not undermine our findings. However, as the reviewer point out, any quantitative attempt in deducing N₂/O₂ to minimize the air influence was invalid then. Thus, we removed the N₂/O₂ deduction data from **Figure 3d**.

In our mass spectra, we indeed detected signals of D⁺, HD, D₂ (m/z=2, 3 and 4) (**Figure R4a** and **Supplementary File 1**) and ¹⁶O¹⁸O, ¹⁸O₂ (m/z=34 and 36) (**Figure R4c** and **Supplementary File 2**), respectively, which confirmed that water decomposition occurred. We are providing the TIC chromatogram and mass spectra in clear photographic form (**Figure R4**), reinforcing the fact that these gases were present in the sample. It's important to reiterate that our methodology is backed by rigorous studies that have employed similar GC-MS analyses for water decomposition products (1-10). The inevitable introduction of air and back-reaction does not invalidate these analyses with isotopically-labeled products, especially ¹⁸O₂ signal (m/z=36) that is not affected by any possible backgrounds.

In summary, according to the reviewer's suggestion, we have provided all the GC-MS raw data (including CSV files) and removed the unscientific N₂/O₂ deduction data in our revised manuscript. Please see **Figure 3d**, **Supplementary Figure S19-20** and **Supplementary Files 1-3**. We believe that the provision of the requested raw data and our detailed explanation alleviate any concerns. We again appreciate the reviewer's insightful and professional comments that have substantially improved the quality and precision of our manuscript.

Figure R4. Raw data figures of (a) mass spectrum for D₂O sample (inset shows the reported figure section in the manuscript), retention time = 1.492 min. (b) TIC chromatogram for D₂O sample. (c) Mass spectrum for ¹⁸O-labeled H₂¹⁸O sample (inset shows the reported figure section in the manuscript), retention time = 2.629 min. (d) TIC chromatogram for ¹⁸O-labeled H₂¹⁸O sample. Detailed CSV matrix see Supplementary Files 1-2.

Figure R5. Raw data figures of (a) mass spectrum for pure N₂ sample (inset shows the reported figure section in the supplementary information), showing no He interference at retention time = 1.423 min. (b) TIC chromatogram for pure N₂ sample. Detailed CSV matrix see Supplementary File 3.

References:

1. Ning, X. et al. Inhibition of CdS photocorrosion by Al₂O₃ shell for highly stable photocatalytic overall water splitting under visible light irradiation. *Appl. Catal. B.* 226, 373-383 (2018).
2. Li, Y. et al. LaOCl-coupled polymeric carbon nitride for overall water splitting through a one-photon excitation pathway. *Angew. Chem. Int. Ed.* 59, 20919-20923 (2020).
3. Huang, Y. et al. Pt Atoms/Clusters on Ni-phytate-sensitized carbon nitride for enhanced NIR-light-driven overall water splitting beyond 800 nm. *Angew. Chem. Int. Ed.* 61, e202212234 (2022).
5. Wang, J. et al. Mn-Doped g-C₃N₄ nanoribbon for efficient visible-light photocatalytic water splitting coupling with methylene blue degradation. *ACS Sustainable Chem. Eng.* 6, 8754-8761 (2018).
6. Wang, L. et al. Van der Waals heterostructures comprised of ultrathin polymer nanosheets for efficient Z-scheme overall water splitting. *Angew. Chem. Int. Ed.* 57, 3454-3458 (2018).
7. Wu, Q. et al. Metal-free Catalyst with large carbon defects for efficiently direct overall water splitting in air at room pressure. *ACS Appl. Mater. Inter.* 12, 20280-30288 (2020).
8. Zhang, et al. Internal quantum efficiency higher than 100% achieved by combining doping and quantum effects for photocatalytic overall water splitting. *Nat. Energy* 8, 504-514 (2023).
9. Wu, J. et al. Breaking through water-splitting bottlenecks over carbon nitride with fluorination. *Nat. Commun.* 13, 6999 (2022).
10. Jiang, E. et al. Vertical growth of O-Vacancy rich LDH atomic layers as OER sensitive reactive sites to boost overall water splitting on perovskite oxides. *ACS Sustainable Chem. Eng.* 10, 16335-16343 (2022).
11. Jiang, E. et al. Nanochannel-induced efficient water splitting at the superhydrophobic interface. *ACS Nano* 17, 10774-10782 (2023).

To Reviewer 5:

Comments:

This work provides a new systematical analysis for the performance of the :PDI²⁻/PDI²⁻ catalyst for visible-light driven overall water splitting process from the photochemistry insights, which is very nice. However, there are still some details to be concerned.

Q1: *For the PDI²⁻ quinoid isomer, the singlet photoluminescence emissions at 540 nm and 585 nm, shown in Fig. 2d, while the corresponding fluorescence peaks are at 646 nm and 690 nm (Fig.2e) via tautomerism transformation. What makes the differences?*

Response: We appreciate the reviewer's comment. Briefly, it was the introduction of hydrazine hydrate into the system for preparation of :PDI²⁻/PDI²⁻ that makes the PL of :PDI²⁻/PDI²⁻ in DMF solution significantly redshift from 540 nm and 585 nm to 646 nm and 690 nm. Namely, the fluorescence feature of PDI²⁻ quinoid

isomer is significantly affected by the presence of hydrazine hydrate as well as its oxidized products. To clarify, let's first consider the individual luminescence spectra of three species each, namely, PDI, :PDI^{2-} diradical, and its tautomer PDI^{2-} , in DMF solutions (refer to Figure R6a-c). The precursor PDI in DMF has three PL peaks (at 542, 577 and 630 nm) (Figure R6a). The isolated reduced PDI^{2-} quinoid isomer has two PL peaks at 540 nm and 585 nm (Figure R6c). However, the triplet diradical :PDI^{2-} of the tautomer $\text{:PDI}^{2-}/\text{PDI}^{2-}$ mixture is non-radiative throughout (Figure R6a). A notable difference exists in the PL spectra between PDI and the reduced tautomer $\text{:PDI}^{2-}/\text{PDI}^{2-}$ mixture. Moreover, the PL characteristics of PDI^{2-} in different solutions can be significantly influenced by the co-existence of other compounds, such as hydrazine hydrate or its oxidized products. Thus, it's understandable that the addition of hydrazine hydrate causes the peaks of PDI^{2-} to redshift significantly from 540 nm and 585 nm to 641 nm and 688 nm, respectively (see Figure R6b). In other words, the presence of extra hydrazine hydrate or its oxidized products significantly affects the fluorescence features of the $\text{:PDI}^{2-}/\text{PDI}^{2-}$ tautomer mixture in solution. In our initial response, we compared the PL emission spectra of $\text{:PDI}^{2-}/\text{PDI}^{2-}$ after reduction with residual hydrazine hydrate and isolated $\text{:PDI}^{2-}/\text{PDI}^{2-}$ without hydrazine hydrate. We attributed the fluorescence changes to the presence of hydrazine hydrate in solution.

To further validate this, we performed additional control experiments. We added varying amounts (0~0.35 μL) of hydrazine hydrate to the DMF solution of the isolated $\text{:PDI}^{2-}/\text{PDI}^{2-}$ tautomer mixture, which was separated from the original reduction system of PDI with hydrazine hydrate by heating and evaporating the solvent in an oxygen-free environment. As shown in Figure R6d, we observed two distinct peaks at 540 nm and 585 nm-attributed to the fluorescence emission of PDI^{2-} -before adding hydrazine hydrate on the isolated $\text{:PDI}^{2-}/\text{PDI}^{2-}$ sample in DMF. A gradual redshift in the PL emission of $\text{:PDI}^{2-}/\text{PDI}^{2-}$ was observed with the addition of hydrazine hydrate. This strongly supports that the presence of hydrazine hydrate in solution indeed caused a significant redshift in the fluorescence of $\text{:PDI}^{2-}/\text{PDI}^{2-}$. This outcome may be due to the coordination interaction between the alkaline hydrazine hydrate molecules and the carboxyl groups of the $\text{:PDI}^{2-}/\text{PDI}^{2-}$ tautomer molecules, a phenomenon that is a common event (1, 2). Further, when we introduced other alkaline small molecules such as ammonia, DEA, and TEOA into the DMF solution of $\text{:PDI}^{2-}/\text{PDI}^{2-}$, we observed similar significant changes and a redshift in the PL emission of $\text{:PDI}^{2-}/\text{PDI}^{2-}$ (See Figure R6e). This supports the idea that coordination interaction causes changes in the fluorescence of $\text{:PDI}^{2-}/\text{PDI}^{2-}$ molecules. We hope these additional experimental details and references provide a clearer understanding of this process. Additional experimental results have been added to the revised manuscript, please see Page 6, Lines 26~32, and Supplementary Fig. 10.

Figure R6. (a) In-situ PL spectra ($\lambda_{\text{ex}}=450$ nm) tracking the emission of PDI after the $2e^-$ reduction by hydrazine hydrate in DMF solution in 30 min. With the addition of $0.25 \mu\text{L}$ hydrazine hydrate (85%), PDI was first converted to non-fluorescent $\text{:PDI}^{\cdot-}$ diradicals. A tautomeric equilibrium from the non-fluorescent diradicals to fluorescent quinoids was then achieved in 30 min. (b) Comparing PL spectra ($\lambda_{\text{ex}}=450$ nm) between pristine PDI and $\text{:PDI}^{\cdot-}/\text{PDI}^{\cdot-}$ samples from in-situ $2e^-$ reduction of PDI for 30 min before and after removing residue hydrazine hydrate in DMF solutions. (c) The PL spectrum ($\lambda_{\text{ex}}=450$ nm) of $\text{:PDI}^{\cdot-}/\text{PDI}^{\cdot-}$ solid precursor sample. (d) PL spectra ($\lambda_{\text{ex}}=450$ nm) of isolated $\text{:PDI}^{\cdot-}/\text{PDI}^{\cdot-}$ DMF solution in addition of different amount of hydrazine hydrate. (e) PL spectrum ($\lambda_{\text{ex}}=450$ nm) spectra ($\lambda_{\text{ex}}=450$ nm) of isolated $\text{:PDI}^{\cdot-}/\text{PDI}^{\cdot-}$ DMF solution in addition of $0.35 \mu\text{L}$ ammonia, DEA, and TEOA.

References:

1. Yersin, H. et al. The triplet state of organo-transition metal compounds. Triplet harvesting and singlet harvesting for efficient OLEDs. *Coord. Chem. Rev.* 225, 2622-2652 (2011).
2. Zhang, G. et al. A dual-emissive-materials design concept enables tumor hypoxia imaging. *Nat. Mat.* 8, 747-751 (2009).

Q2: *Why could the Stokes shift of UV-vis absorption and PL emission spectra suggest the lower reorganization energy, please give some references.*

Response: We appreciate the reviewer's comment. The Stokes shift, or the difference in energy between the maxima of the absorption and emission spectra, is a reflection of the difference in equilibrium geometry between the ground state and the excited state of a molecule. When a molecule absorbs a photon and transitions from the ground state to the excited state, it may undergo structural reorganization to reach a new equilibrium geometry. This reorganization requires energy, which is referred to as the reorganization energy.

When the Stokes shift is small, it implies that the absorption and emission maxima are close together. This, in turn, suggests that the equilibrium geometries of the ground and excited states are quite similar. Thus, the structural reorganization required for the molecule to transition between these states is minimal, indicating a lower reorganization energy.

This relationship between Stokes shift and reorganization energy is a fundamental concept in the field of photo-physics and is discussed in depth in the following references:

1. Marcus, R. A. Electron transfer reactions in chemistry: Theory and experiment (Nobel lecture). *Angew. Chem. Int. Ed.* 32, 1111-1121 (1993).
2. Lakowicz, J. R. Principles of fluorescence spectroscopy. Springer, (2006).
3. Pugžlys, A. et al. Temperature-dependent relaxation of excitons in tubular molecular aggregates: Fluorescence decay and Stokes shift. *J. Phys. Chem. B.* 110, 20268-20276 (2006).
4. May, V. and Kühn, O. Charge and energy transfer dynamics in molecular systems. Wiley, (2011).
5. Kashani, S., et al. Relating reorganization energies, exciton diffusion length and non-radiative recombination to the room temperature UV-vis absorption spectra of NF-SMA. *Mater. Horiz.* 10, 443-453 (2023).

Q3: *How to keep high stability of the :PDI²⁻/PDI²⁻ nanobelt in the acid solution for driving HER?*

Response: The :PDI²⁻/PDI²⁻ that we employ features two carboxyl groups at the terminals on its perylene framework, thereby ensuring its stability even in acidic environments. Its self-assembled structure also demonstrates robustness under acidic conditions since its close-packed construction mainly relies on the strong π - π interaction between perylene core of :PDI²⁻/PDI²⁻ building blocks. In a high-acidity environment, excess protons in the system only drive :PDI²⁻/PDI²⁻ to undergo a reversible transformation to :PDI-H₂/PDI-H₂. To substantiate this point, we conducted HER testing under acidic conditions, affirming that the :PDI²⁻/PDI²⁻ nanobelt continues to operate stably even at a pH of 3.0 (adjusted by diluted HCl solution) (**Figure R7**). This

evidence strongly indicates that our PDI-based system can maintain high stability when driving HER in acidic solutions.

Figure R7. Long-term recycling hydrogen production performances over :PDI²⁻/PDI²⁻ nanobelts (120 h) under white light (1000 mW cm⁻²) in HCl aqueous solution (pH=3.0) for 50 h with 15-min interval continuous vacuum pumping.

Q4: Direct $S_0 \rightarrow T_1$ transition is ascribed to spin-orbit coupling, please give more details or references about the mechanism.

Response: We appreciate the reviewer's query about the mechanism of direct $S_0 \rightarrow T_1$ transition attributed to spin-orbit coupling. Typically, in many molecules, the lowest energy transition is from the singlet ground state (S_0) to the first singlet excited state (S_1). However, under certain conditions in particular molecules, a direct transition from the ground singlet state (S_0) to the first triplet state (T_1) can occur. This is made possible when the spin-orbit coupling is sufficiently strong to facilitate 'spin-flipping' during the electronic transition.

This 'spin-flipping' is generally prohibited by the 'spin-conservation' rule (1). However, spin-orbit coupling allows for an interaction between the spatial distribution of the electron (which determines the electronic state) and its spin state. This interaction can lead to a 'mixing' of singlet and triplet states, enabling transitions that would otherwise be spin-forbidden. The strength of the spin-orbit coupling (and therefore the likelihood of such transitions) increases with the atomic number: the heavier the atom, the stronger the coupling. which is why such transitions are more frequently observed in molecules containing heavy atoms (2-6). In our all-organic spin-hybrid system, the 'mixing' of singlet and triplet states in the diradical-quinoid hybrid bypasses the 'spin-conservation' rule, enables the spin-forbidden direct $S_0 \rightarrow T_1$ excitation.

For now, the direct $S_0 \rightarrow T_1$ excitation is an unusual phenomenon, sporadically found in a minority of molecular systems with spin-coupling. The exact causes of this occurrence are still not fully understood and

are the subject of ongoing research. For a more detailed discussion and reported molecular systems, please refer to the following references:

1. Baryshnikov, G. et al. Theory and calculation of the phosphorescence phenomenon. *Chem. Rev.* **117**, 6500-6537 (2017).
2. Han, S. Y. et al. Lanthanide-doped inorganic nanoparticles turn molecular triplet excitons bright. *Nature* **587**, 594-612 (2020).
3. Nakajima, M. et al. Direct $S_0 \rightarrow T_n$ transition in the photoreaction of heavy-atom-containing molecules. *Angew. Chem. Int. Ed.* **59**, 6847-6852 (2020).
4. Liu, D. Y. et al. Exploit the benefit of excitation in triplet-triplet annihilation upconversion to attain large anti-stokes shifts: tuning the triplet state lifetime of a tris(2, 2'-bipyridine) osmium (II) complex. *Dalton Trans.* **47**, 8619-8628 (2018).
5. Ishii, K., Wada, J. & Murata, K. Direct observation of the $S \rightarrow T$ transition in phosphorescent platinum (II) octaethylporphyrin, evidenced by magnetic circular dichroism. *J. Phys. Chem. Lett.* **11**, 9828-9833 (2021).
6. Ando, A. et al. Aggregation-enhanced direct $S_0 \rightarrow T_n$ transitions and room-temperature phosphorescence in gold(I)-complex single crystals. *Aggregate* **3**, 125-134 (2022).

Q5: *How to evaluate the charge separation degree within the :PDI²⁻/PDI²⁻ organic semiconductor?*

Response: We appreciate the reviewer's inquiry about the evaluation of the charge-separation (CS) degree within the :PDI²⁻/PDI²⁻ organic semiconductor samples. There should be two direct methods usually employed to assess this CS degree.

The first is a direct comparison of the ground state bleach signal and the directly excited state (i.e., the CS state) intensities by the transient absorption (TA) spectra. As illustrated in Figure R8, by contrasting the intensities of the CS state signal and the bleach signal, the internal CS degree under $\lambda_{ex}=540$ nm in the :PDI²⁻/PDI²⁻ before self-assembly and after 120 h of self-assembly are 43.26% and 51.35% respectively.

The second method involves affixing powder materials to an electrode to form a photoconductive device, and by employing a Surface Photovoltage (SPV) test, the sensitivity of the CS degree of samples to different excitation wavelengths can be directly compared. It's important to note that these measurements are specific to the photo-sensitive device itself (due to the weak conductivity between the powders, the CS degree usually will be several orders of magnitude lower than that directly measured from the powder materials by TA), and do not reflect the internal CS population of the powder materials themselves. However, differences in CS at different wavelengths on the assembled device can still be directly compared. As shown in Figure R9, the SPV tests of :PDI²⁻/PDI²⁻ before self-assembly and after 120 h of self-assembly suggest that the self-assembled :PDI²⁻/PDI²⁻ organic semiconductor has a better CS performance at higher wavelengths compared to the non-assembled material.

Figure R8. (a), (c) Femtosecond transient absorption: 2D map and (b), (d) TA kinetic traces extracted under different wavelength of :PDI²⁻/PDI²⁻ (0 h) and :PDI²⁻/PDI²⁻ (120 h). (e) Calculated internal charge separation degree of :PDI²⁻/PDI²⁻ (0 h) and :PDI²⁻/PDI²⁻ (120 h), $\lambda_{ex}=540$ nm.

Figure R9. (a) Surface photovoltage profiles (inset shows the schematic setup for the SPV measurements) of :PDI²⁻/PDI²⁻ (120 h) at different wavelengths. (b) Calculated charge separation degree of SPV devices prepared by :PDI²⁻/PDI²⁻ (0 h) and :PDI²⁻/PDI²⁻ (120 h) at different wavelengths.

Reviewers' comments:

Reviewer #3 (Remarks to the Author):

In this round, the author has focused on responding to the questions I raised in the previous round. First, I would like to thank the authors for their efforts. However, there are still flaws in this response, especially on the issue of isotope traceability, this experiment is not carried out carefully. It brings huge hidden dangers to the performance reliability of the entire article. I pointed them out as follows:

1 The authors claimed that they separate analyte gases before they enter the mass spectrometer and distinct retention times for D₂ (1.492 min) and ¹⁸O₂ (2.629 min) in TIC. But according to the data provided by the authors, this answer is inaccurate. The authors provide two different TIC and attribute them to the D₂ and ¹⁸O₂, and both of them exhibit one peak, how does this prove that the two gases are separated? In addition, if the D₂ and O₂ could be separated before the analyte gases enter the mass spectrometer, the N₂ could also be separated. Unfortunately, both N₂ ($m/z = 28$) and D₂ ($m/z = 4$) can be seen in both mass spectrum. In this regard, I do not consider the data provided by the author to be proof of successful traceability.

2 The reaction of H₂ with O₂ during ionization in the mass spectrometer is unavoidable. Once this reaction occurs, the possible detection of H₂ and D₂ in mass spectrometry is not practical based on our research. Although the author provides the so-called mass spectrum of D₂ ($m/z=2$, $m/z=3$, $m/z=4$), by observing the mass spectrum at different times provided by authors, we can see that this mass-to-charge ratio signal can be seen in both mass spectra, which is enough to show that this signal is only a background signal of the instrument.

Reviewer #5 (Remarks to the Author):

Accept

Reply to reviewers' comments

To Reviewer 3:

Comments:

In this round, the author has focused on responding to the questions I raised in the previous round. First, I would like to thank the authors for their efforts. However, there are still flaws in this response, especially on the issue of isotope traceability, this experiment is not carried out carefully. It brings huge hidden dangers to the performance reliability of the entire article. I pointed them out as follow:

Response: We appreciate the reviewer's meticulous and professional feedback, which pointed out the issues with our initial mass spectrometry data and its uncaredful handling. We have taken the comments very seriously.

As the reviewer noted, our mass spectrometry data collected two years ago did not sufficiently substantiate our conclusion that water-splitting occurred on the catalyst. Despite clear evidence from numerous in-lab online gas chromatography tests, we realize that our previous reporting lacked precision.

To strengthen our work with a solid chain of evidence and to address the reviewer's concerns, we performed new mass spectrometry analyses with the guidance of Dr. Dujuan Yang and Dr. Fengfeng Yang at the Agilent Shared Laboratory (ASL) in Shanghai. With their help, we conducted meticulous comparative experiments with pre-/post-reaction (light-off/light-on) conditions to distinguish trace isotopic products accurately from the inlet background. Additionally, we performed isotopic Extracted Ion Chromatograms (EIC) to analyze the samples further. This approach confirmed an increase in isotopic products post-reaction. The details of these experiments are included in our responses to Q1 and Q2 below. The detailed mass spectrometry experimental testing methods were written in the last part of the response letter and included in the revised manuscript.

We thank the reviewer again for the professionalism and attention to detail, which have significantly improved the data quality of our manuscript.

Q1: *The authors claimed that they separate analyte gases before they enter the mass spectrometer and distinct retention times for D₂ (1.492 min) and ¹⁸O₂ (2.629 min) in TIC. But according to the data provided by the authors, this answer is inaccurate. The authors provide two different TIC and attribute them to the D₂ and ¹⁸O₂, and both of them exhibit one peak, how does this prove that the two gases are separated? In addition, if the D₂ and O₂ could be separated before the analyte gases enter the mass spectrometer, the N₂ could also be separated. Unfortunately, both N₂ (m/z =28) and D₂ (m/z =4) can be seen in both mass spectrum. In this regard, I do not consider the data provided by the author to be proof of successful traceability.*

Response: We appreciate the reviewer's concerns about the Total Ion Chromatogram (TIC) and the Mass Spectra (MS). Upon revisiting our initial TIC and MS analysis, as well as the testing conditions, we recognized that the Agilent HP-5MSUltra Inert column we used, was not effective in separating hydrogen from oxygen/nitrogen. This inadequacy led to a misrepresentation of the true state of the separated gases in our previous mass spectra. Repetitive experiments using the same column at the Agilent Shared Laboratory (ASL) yielded similar results (**Figure R1**), confirming that our earlier MS analytical method was unsuitable for our system, just as the reviewer suspected, particularly for analyzing deuterium isotopic products.

Figure R1. GC-MS analytical data using Agilent HP-5MSUltra Inert column at 70 eV for standard gas: the TIC of (a) standard H₂ (1 mL) and (b) standard D₂ (1 mL).

To address this, we have since employed an Agilent CP-Molsieve 5A column, which is the best candidate that the Agilent company has been recommending for efficiently separating hydrogen, oxygen, and nitrogen. At 70 eV ionization energies, He and D₂ will be ionized simultaneously, which cannot avoid the influence of carrier gas He on mass spectrometry. At 20 eV ionization energies, He cannot be ionized, only D₂ is ionized. Therefore, we have validated this separation by introducing pure H₂ and D₂ gases, as shown in **Figure R2** (upon 20 eV) and **Figure R3** (upon 70 eV). It should also be stressed because GC-MS sampling/injection is an ex-situ process, a segment of air from the sampling port is inevitably drawn into the system and vented back after each test. Consequently, N₂ and O₂ signals can still be observed even when the sample is pure H₂/D₂. At 20 eV, the mass spectral signal for H₂ ($m/z=2$) is virtually undetectable in pure hydrogen samples (**Figure R2c**), whereas the D₂ signal ($m/z=4.2$) and some D₃⁺ fragments ($m/z=6.2$) are observed in pure deuterium samples. This indicates that the current mass spectrometer can discern D₂ from the carrier gas signal since He has absolutely no fragment signal of $m/z=6.2$. However, H₂ ionization and detection at 20 eV ionization conditions is not feasible and the ionization potential has to be increased to 70 eV to meet the requirement of

determination of H₂. At 70 eV, due to the use of He as the carrier gas, a background peak of He (m/z=4.2) inevitably interferes with the identification of D₂. Nevertheless, the D₃⁺ fragment signal at m/z=6.2 remains a reliable isotopic marker for D₂. Signals for H₂O (m/z=18.1) and D₂O (m/z=20.1) were also detected at both 20 eV and 70 eV, indicating the presence of back-reaction between H₂/D₂ and O₂ in the ionization process of MS determination, and concurrently confirming the purity of our H₂ and D₂ standard gases (predominantly D₂O at m/z=20.1 in D₂ standard gas). Despite these complexities, signals at m/z=4.2 and m/z=6.2 were still observable. More importantly, Extracted Ion Chromatogram (EIC) analysis was performed on the sample at 20 eV, and it was observed that with the same He acting as carrier gas, the standard H₂ sample had no signal at m/z=4.2 (D₂) (**Figure R2e**), whereas it appeared when standard D₂ was used as the testing sample (**Figure R2f**), proving the reliability of the data extracted at m/z=4.2 at 20 eV. Moreover, due to the He carrier gas, we conducted EIC on the sample at 70 eV and found that there was no signal from the standard H₂ gas at m/z=6.2 (D₃⁺) window (**Figure R3e**), instead, it appeared in the D₂ testing sample case as expected (**Figure R3f**), proving the reliability of the data extracted at m/z=6.2, 70 eV.

Figure R2. GC-MS analytical data using Agilent CP-Molsieve 5A column at 20 eV for standard gas: the TIC of (a) standard H₂ (1 mL) and (b) standard D₂ (1 mL), the corresponding MS spectra (RT=1.167 min) of (c) standard H₂ (1 mL) and (d) standard D₂ (1 mL), and the EIC (m/z=4.2) of (e) standard H₂ (1 mL) and (f) standard D₂ (1 mL).

In summary, through comparable control experiments with standard H₂ and D₂ at 20 eV and 70 eV, respectively, we have identified reliable isotopic characteristic signal markers for tracing D₂ gas, specifically at (m/z=4.2, 20 eV) and (m/z=6.2, 70 eV). Based on the isotopic signals of these two markers, we conducted isotopic EIC analysis on our samples, namely catalyst+H₂O/D₂O and catalyst+H₂O/H₂¹⁸O, and fully verified

the occurrence of water-splitting reaction. The specific results are shown in the response of Q2.

Figure R3. GC-MS analytical data using Agilent CP-Molsieve 5A column at 70 eV for standard gas: the TIC of (a) standard H₂ (1 mL) and (b) standard D₂ (1 mL), the corresponding MS spectra (RT=2.100 min) of (c) standard H₂ (1 mL) and (d) standard D₂ (1 mL), and the EIC (m/z=6.2) of (e) standard H₂ (1 mL) and (f) standard D₂ (1 mL).

Q2: *The reaction of H₂ with O₂ during ionization in the mass spectrometer is unavoidable. Once this reaction occurs, the possible detection of H₂ and D₂ in mass spectrometry is not practical based on our research. Although the author provides the so-called mass spectrum of D₂ (m/z=2, m/z=3, m/z=4), by observing the mass spectrum at different times provided by authors, we can see that this mass-to-charge ratio signal can be seen in both mass spectra, which is enough to show that this signal is only a background signal of the instrument.*

Response: We very appreciate the reviewer’s professional feedback. In response to the concerns regarding detecting H₂ and D₂ in mass spectrometry, we conducted the isotopic EIC to further analyze the samples (catalyst+H₂O/D₂O) before/after reaction.

As illustrated in **Figure R4**, at an ionization potential of 20 eV, a new signal attributed to D₂ was detected in the sample after reaction Total Ion Chromatogram (TIC) at a retention time (RT) of 1.257 minutes. Given that the reaction was conducted under a nitrogen (N₂) atmosphere, the partial pressure of the newly generated D₂ was rather low; hence, the signal required a 210-fold amplification to be visible with an identical y-axis scale (**Figure R4a, b**). This pre- and post-reaction comparison convincingly demonstrates the formation of H₂/D₂. To further corroborate that the detected D₂ was indeed isotopically substituted, we extracted the

$m/z=4.2$ ion chromatogram from the TIC profile. This analysis revealed that the $m/z=4.2$ extraction chromatogram peaked exclusively at $RT=1.257$, while no signals present during the retention times for O_2 and N_2 , which also effectively rules out ionization of carrier gas He as a contributing factor (Figure R4c, d). Therefore, the newly emerged signal at $RT=1.257$ minutes for the sample of post-reaction represents unambiguously the D_2 gas, confirming the catalytic decomposition of D_2O .

Figure R4. GC-MS analytical data using Agilent CP-Molsieve 5A column at 20 eV for D isotope-tracing experiments in D_2O overall splitting using catalysts (catalyst/ $D_2O=20$ mg/10 mL): the TIC of 2 mL products generated by (a) light-off and (b) light-on for 2 h (inset shows the 210-fold amplification of y-axis scale), the EIC ($m/z=4.2$) of 2 mL products generated by (c) light-off and (d) light-on for 2 h.

Additionally, the experiment was replicated using an ionization energy of 70 eV (employing a longer column of the same type to enhance the separation of H_2 , O_2 , and N_2 signals), as depicted in Figure R5. The formation of D_2 is distinctly identifiable in the TIC at $RT=2.16$ minutes. However, due to the interference from the He background at $m/z=4.2$, only D_3 fragments at $m/z=6.2$ can be acknowledged as indicative of D_2 gas presence (Figure R5c, d). We monitored the $m/z=6.2$ extracted ion chromatogram (EIC) from the TIC profile (Figure R5f), and the integration curve for the D_3 fragment ($m/z=6.2$) extraction emerged in the post-reaction sample, identical to the $m/z=4.2$ EIC signal profile observed under the 20 eV ionization condition (Figure R4). On the contrary, it did not occur for the pre-reaction sample case (Figure R5e). Meanwhile, we monitored the $m/z=6.2$ EIC from the TIC profile on the unlabeled Catalyst+ H_2O sample and did not extract any D_3 fragment signals (Figure R8e, f). These comparative analyses robustly validate our findings that D_2O indeed undergoes decomposition on the catalyst under illumination, resulting in the production of D_2 .

Figure R5. GC-MS analytical data using Agilent CP-Molsieve 5A column at 70 eV for D isotope-tracing experiments in D_2O overall splitting using catalysts (catalyst/ D_2O =20 mg/10 mL): the TIC of 2 mL products generated by (a) light-off and (b) light-on for 2 h (inset shows the 130-fold amplification of y-axis scale), the corresponding MS spectra (RT=2.166 min) of 2 mL products generated by (c) light-off and (d) light-on for 2 h (inset shows the 400-fold amplification of y-axis scale), EIC ($m/z=6.2$) of 2 mL products generated by (e) light-off and (f) light-on for 2 h.

Furthermore, we conducted isotopic substitution experiments with $H_2^{18}O$ to confirm further the water-splitting by the production of oxygen (Figure R6 and Figure R7). While the increase in O_2 in the Total Ion Chromatogram (TIC) was indiscernible due to the presence of the air inlet background, the extraction of the $m/z=36$ signal at both 20 eV and 70 eV ionization potential indicated a significant increase in the characteristic signal of $^{18}O_2$ in the post-reaction samples. For the unlabeled H_2O sample, we found that EIC ($m/z=36.0$) did not change after illumination (Figure R8g, h). This uptick corresponds to the formation of $^{18}O_2$, substantiating our conclusions.

Figure R6. GC-MS analytical data using Agilent CP-Molsieve 5A column at 20 eV for ^{18}O isotope-tracing experiments in H_2^{18}O overall splitting using catalysts (catalyst/ H_2^{18}O =20 mg/10 mL): the TIC of 1 mL products generated by (a) light-off and (b) light-on for 2 h, the EIC ($m/z=36.0$) of 1 mL products generated by (c) light-off and (d) light-on for 2 h.

Figure R7. GC-MS analytical data using Agilent CP-Molsieve 5A column at 70 eV for ^{18}O isotope-tracing experiments in H_2^{18}O overall splitting using catalysts (catalyst/ H_2^{18}O =20 mg/10 mL): the TIC of 1 mL products generated by (a) light-off and (b) light-on for 2 h, the corresponding MS spectra (RT=2.689 min) of 1 mL products generated by (c) light-off and (d) light-on for 2 h, EIC ($m/z=36.0$) of 1 mL products generated by (e) light-off and (f) light-on for 2 h.

Figure R8. GC-MS analytical data using Agilent CP-Molsieve 5A column at 70 eV for overall water splitting using catalysts (catalyst/H₂O =20 mg/10 mL): the TIC of 2 mL products generated by (a) light-off and (b) light-on for 2 h (inset shows the 70-fold amplification of y-axis scale), the corresponding MS spectra (RT=2.138 min) of 2 mL products generated by (c) light-off and (d) light-on for 2 h (inset shows the 2000-fold amplification of y-axis scale), EIC (m/z=6.2) of 2 mL products generated by (e) light-off and (f) light-on for 2 h, EIC (m/z=36.0) of 2 mL products generated by (g) light-off and (h) light-on for 2 h.

In summary, our new GC-MS results coupled with additional Extracted Ion Chromatogram (EIC) analyses, have successfully traced the generation of water splitting products. Explicitly addressing the concerns raised by the reviewer regarding potential interferences of helium carrier gas in the detection of D₂, we have eliminated such interferences through meticulous comparative experiments and EIC analyses. All of these experimental results are added in the revised supplementary information (**Supplementary Fig. 18**. And **19**) and prompted in the revised text. We hope our supplementary experimental results address and resolve any doubts the reviewer may have had about our findings. We thank the reviewer again for the help in improving the data quality and our manuscript.

Detailed isotopically labeled measurements

Preparation of isotopically labeled samples

In the Nitrogen glove box, 20 mg :PDI²⁻/PDI²⁻ nanobelts was dispersed uniformly in 10 mL of D₂O (H₂¹⁸O or H₂O) and transferred into a 50 mL volume vial. The gas (5 mL) was collected by the injection needle with a throttle valve into sealed gas bag, denoted as Light-off sample. The vials were sealed and irradiated using a 300 W Xe-lamp (1000 mW cm⁻², CEL-HXF300-T3, Beijing China Education Au-light Technology Co. Ltd, China) for 2 hours. The reaction gas (5 mL) was collected by the injection needle with a throttle valve into sealed gas bag, denoted as Light-on sample. The Light-off samples and Light-on samples were collected from the same vial.

Using Agilent CP-Molsieve 5A as a column for isotope-labeled samples analysis

The 2 mL gas samples were collected and manually injected by gas-tight syringes (the FTFE luer lok (PN5190-1534), 0~2.5 mL) and then analyzed by gas chromatography-mass spectrometry (8890-5977B GC-MS instrument, Agilent Technologies, USA) equipped with an Agilent CP-Molsieve 5A column (25 m × 250 μm × 30 μm, Agilent Technologies, USA) in GC-MS. Helium was used as carrier gas. The column was maintained at 100 °C for 5 min, and the flow of the carrier was 1.5 mL min⁻¹. The temperatures of the injector and MS source were set to be 100 °C and 230 °C, respectively. The ionization potentials used for analysis were set to 20 eV and 70 eV, respectively.

Figure R9. Photograph of 8890-5977B GC-MS instrument in Agilent Shared Laboratory (ASL, Shanghai).

REVIEWER COMMENTS

Reviewer #3 (Remarks to the Author):

I would like to thank the authors for their efforts again. In this round of revisions, I acknowledge that the author's proposed method for establishing source tracing mass spectrometry is sound. Most of the data provided by the author are reasonable.

However, there are still some loopholes in this experiment of isotope traceability, and the authors need to focus on it.

1. The experiment of isotope traceability were carried out on the off-line reaction system. Although the gas was transfer via a gas bag and a injection needle with a throttle valve, the Leakage during transport is unavoidable, especially for Luer-Lok, the needle can be sealed, but the pillow cannot be sealed. Most importantly, although the authors confirm the production of $^{18}\text{O}_2$ and D_2 in the system, they cannot prove that this is the main product due to the off-line reaction system was employed. The author are strongly suggest to carried out the experiment on the on-line reaction system or headspace vials. If using headspace vials, the headspace sampler could rule out the interference from oxygen in the air; If using on-line reaction system (a vacuum environment is provided by a molecular pump), The same vacuum level as gas-mass spectrometry allows for the elimination of air interference during the tandem reactor with gas-mass spectrometry.

2. I cannot understand that why the D_2 could be detected in MS as the form of D_2^+ ($m/z=4$) and D_3^+ ($m/z=6$), but H_2 cannot be detected as the as the form of H_2^+ ($m/z=2$) and H_3^+ ($m/z=3$)? Under the same conditions provided by the authors, the mass spectra of these are not proportional.

Reply to reviewers' comments

To Reviewer 3:

Comments:

I would like to thank the authors for their efforts again. In this round of revisions, I acknowledge that the authors proposed method for establishing source tracing mass spectrometry is sound. Most of the data provided by the author are reasonable. However, there are still some loopholes in this experiment of isotope traceability, and the authors need to focus on it.

***Q1:** The experiment of isotope traceability were carried out on the off-line reaction system. Although the gas was transfer via agas bag and a injection needle with a throttle valve, the Leakage during transport is unavoidable, especially for Luer-Lok, the needle can be sealed, but the pillow cannot be sealed. Most importantly, although the authors confirm the production of $^{18}\text{O}_2$ and D_2 in the system, they cannot prove that this is the main product due to the off-line reaction system was employed. The author are strongly suggest to carried out the experiment on the on-line reaction system or headspace vials. If using headspace vials the headspace sampler could rule out the interference from oxygen in the air; If using on-line reaction system (a vacuum environment is provided by a molecular pump), The same vacuum level as gas-mass spectrometry allows for the elimination of air interference during the tandem reactor with gas-mass spectrometry.*

Response: We appreciate the reviewer's meticulous and professional feedback, which concerns our off-line mass spectrometric measurement method, especially with gas-bags. To mitigate these issues the reviewer raised, we have established an enhanced headspace sampling system driven by high-pressure N_2 flow, cordially facilitated by the Agilent Shared Laboratory (ASL) in Shanghai. The detailed configuration of our new setup is illustrated in Figure R1.

We performed rigorous isotopic GC-MS characterizations using this system, employing headspace vials to effectively rule out atmospheric oxygen interference. The control experimental results, as demonstrated in Figure R2, confirm the absence of H_2 signal ($m/z = 2$ or 3) at the same signal intensity level for pure H_2 and D_2 . However, the $m/z=6.1$ (D_3^+) signal still acted as a reliable marker for D_2 . With the pre-post reaction comparison, a significant increase in the $m/z=6.1$ signal post-reaction in both GC-MS and EIC (Extracted Ion Current) traces verifies the production of D_2 , as the EIC signal aligns perfectly with the TIC (Total Ion Current) peak position (Figure R3).

In addition, our ^{18}O isotope-tracing experiments with H_2^{18}O have similarly produced compelling evidence (Figure R4). We observed a pronounced enhancement of the $m/z=36$ signal post-reaction, which was virtually non-existent before the reaction. This finding decisively addresses previous concerns regarding isotopic contamination from the atmosphere, as it contrasts sharply with our earlier off-line experiments using gas bags.

In summary, the newly conducted headspace GC-MS experiments have not only eliminated the potential for atmospheric isotopic contamination, but have also significantly improved the reliability and validity of our findings. We are confident that the measures we have taken thoroughly address the concerns the reviewer have raised. We are grateful for the critical role the reviewer's expert review in guiding us to enhance the quality of our research.

Figure R1. Photograph of GC-MS instrument, headspace injection device and headspace vial with our catalyst.

Figure R2. GC-MS analytical data under 70 eV for standard gas using headspace vial (sampled through drainage method): the TIC of standard H₂ and standard D₂, and the corresponding MS spectra (RT=2.493 min) of (c) standard H₂ and (d) standard D₂.

Figure R3. GC-MS analytical data using headspace vial under 70 eV for D isotope-tracing experiments in D_2O overall splitting using our catalysts (catalyst/ D_2O =20 mg/2 mL): the TIC of products generated by (a) light-off and (b) light-on for 10 h, the corresponding MS spectra (RT=2.493 min) of products generated by (c) light-off and (d) light-on for 10 h, EIC ($m/z=6.1$) of products generated by (e) light-off and (f) light-on for 10 h.

Figure R4. GC-MS analytical data using headspace vial under 70 eV for ^{18}O isotope-tracing experiments in $H_2^{18}O$ overall splitting using catalysts (catalyst/ $H_2^{18}O$ =20 mg/2 mL): the TIC of products generated by (a) light-off and (b) light-on for 2 h, the corresponding MS spectra (RT=3.066 min) of products generated by (c) light-off and (d) light-on for 2 h, EIC ($m/z=36.0$) of products generated by (e) light-off and (f) light-on for 2 h.

Q2: *I cannot understand that why the D₂ could be detected in MS as the form of D²⁺ (m/z=4) and D³⁺ (m/z=6), but H₂ cannot be detected as the as the form of H²⁺ (m/z=2) and H³⁺ (m/z=3)? Under the same conditions provided by the authors, the mass spectra of these are not proportional.*

Response: We appreciate the reviewer's insightful comments regarding the detection of H₂⁺ (m/z=2) and H₃⁺ (m/z=3) versus D₂⁺ (m/z=4) and D₃⁺ (m/z=6) in our mass spectrometry (MS) analysis. We have engaged in thorough discussions with Agilent company and several mass spectrometry engineers and conducted a series of exploratory control experiments to address the reviewer's question.

We began by repeatedly testing H₂ standard samples at both 18 eV and 70 eV ionization energies. Despite the high concentration of standard H₂ gas, both the TIC and MS signals were weaker compared to standard D₂ gas samples. Notably, the H₂⁺ (m/z=2) and H₃⁺ (m/z=3) MS signals were nearly undetectable at 70 eV (see **Figure R5a**), while D₂ showed a clear D₃⁺ (m/z=6) fragment signal under the same conditions (see **Figure R2**), consistent with our previous results using gasbag samples. We hypothesize that the m/z=6 signal detected under D₂ gas test conditions originates from the formation of protonated D₃⁺ fragments following ionization. These signals are observable at both 70 eV and 20 eV. In contrast, under the same conditions, the H₃⁺ (m/z=3) signal for H₂ was very faint (see **Figures R5a** and **R5b**), likely due to differences in instrument detection limits for m/z ratios and the distinct pathways of H⁺/D⁺ rearrangement reactions. When we reduced the ionization energy to 18 eV and amplified the MS signal by 200 times, we could observe extremely weak m/z=2 and m/z=3 signals (see **Figure R5d**), similar to the standard D₂ gas sample results at 20 eV (see **Figure R5e**). Although trace m/z=2 and m/z=3 signals were detected, their TIC and MS signal intensities were still negligible, 2 to 3 orders of magnitude smaller compared to D₂ signals at the same or even lower concentrations (see **Figures R5d** and **R5e**). These results indicate that our GC-MS method has limited detection sensitivity for signals with m/z ≤ 3. To conclusively verify the presence of H₂ molecules in our hydrogen gas standard samples, we utilized the Selected Ion Monitoring (SIM) mode and successfully captured a signal at m/z=2, unequivocally confirming the existence of H₂⁺ ions in our samples (see **Figures R5f** and **R5g**). This substantiates our presumption that the inability to detect the m/z=2 signal in scan mode is attributable to the limitations of the detector's sensitivity.

We speculate that during the ionization process, H₂ and D₂ are ionized into H⁺ and D⁺ fragments, respectively. Although there are slight differences in dissociation energies between H₂ and D₂, the efficiency of ionization and rearrangement is mainly affected by the kinetic isotope effect (KIE) due to differences in vibrational energy levels. This results in D₃⁺ being observable at 70 eV while H₃⁺ can only be detected at 18 eV. More importantly, while our GC-MS mass spectrometer theoretically has a detection limit down to m/z=1.6, thereby precluding the detection of H⁺ fragments, signals with m/z ratios ≤ 3 such as D⁺, H₂⁺, and H₃⁺ are also challenging to detect unequivocally due to detection limit and sensitivity issues on our single

quadrupole (SQ) GC-MS system. These factors collectively contribute to the faint signals for H_3^+ ($m/z=3$). Nevertheless, our meticulous control experiments with isotopic substitution, compared to D_2 standard gas, have conclusively verified the production of D_2 , which sufficiently supports the conclusions of our paper. We are grateful for the valuable comments offered by the reviewer again.

Figure R5. GC-MS analytical data for standard H_2 : the TIC of standard H_2 under (a) 70 eV and (b) 18 eV using headspace vial (sample through drainage method), the corresponding MS spectra (RT=2.493 min) under (c) 70 eV and (d) 18 eV (below: the 160-fold amplification of y-axis scale), (e) the MS spectra of standard D_2 under 20 eV using injector, (f) the TIC of standard H_2 under 18 eV with SIM model using headspace vial (sampled through drainage method), (g) the corresponding MS spectra (RT=2.496 min).

REVIEWER COMMENTS

Reviewer #3 (Remarks to the Author):

The authors addressed almost all of concerns that I raised. But before accepted for publication, the authors should explain why there is no peak for H₂ in TIC (Fig. R4b) when using H₂¹⁸O as the water source for O₂ detection?

Reply to reviewers' comments

To Reviewer 3:

Comments:

The authors addressed almost all of concerns that I raised. But before accepted for publication, the authors should explain why there is no peak for H₂ in TIC (Fig. R4b) when using H₂¹⁸O as the water source for O₂ detection?

Response: We appreciate the reviewer's positive feedback. We think the absence of the H₂ peak in the TIC profile of the H₂¹⁸O-substituted experiment is attributed to the low detection sensitivity and inadequate light irradiation time (2 h).

To further address the reviewer's concern, we performed head-space H₂¹⁸O-substituted GC-MS measurements on samples with prolonged irradiation times (5-48 h). As shown in **Figure R1**, with light-on for 5-48 h, a visible H₂ peak (with 260-fold amplification) emerged and increased with the irradiation time in TIC profiles. Meanwhile, the peak of ¹⁸O₂ in EIC profiles increased accordingly. Due to the instrument's low sensitivity to H₂, the peak for H₂ can only be observed in TIC when the concentration of H₂ in the sample reaches a sufficiently high level. We are grateful again for the reviewer's expert review.

Figure R1. GC-MS analytical data using headspace vial under 70 eV for ¹⁸O isotope-tracing experiments in H₂¹⁸O overall splitting using catalysts (catalyst/H₂¹⁸O = 20 mg/2 mL): the TIC of products generated by (a) light-on for 5 h, (b) light-on for 10 h and (c) light-on for 48 h (inset shows the 260-fold amplification of y-axis scale), EIC (m/z=36.0) of products generated by (d) light-on for 5 h, (e) light-on for 10 h and (f) light-on for 48 h.